# Actuating compact wearable augmented reality devices by multifunctional artificial muscle

Dongjin Kim[1,6], Baekgyeom Kim [1,6], Bongsu Shin[2,3,6], Dongwook Shin[1], Chang-Kun Lee[2,3], Jae-Seung Chung[2,3], Juwon Seo[2,3], Yun-Tae Kim[2,3], Geeyoung Sung[2,3], Wontaek Seo[2], Sunil Kim[2], Sunghoon Hong[2], Sungwoo Hwang [2,4], Seungyong Han [1✉], Daeshik Kang [1✉], Hong-Seok Lee [2,5✉] & Je-Sung Koh [1✉]

An artificial muscle actuator resolves practical engineering problems in compact wearable devices, which are limited to conventional actuators such as electromagnetic actuators. Abstracting the fundamental advantages of an artificial muscle actuator provides a small-scale, high-power actuating system with a sensing capability for developing varifocal augmented reality glasses and naturally fit haptic gloves. Here, we design a shape memory alloy-based lightweight and high-power artificial muscle actuator, the so-called compliant amplified shape memory alloy actuator. Despite its light weight (0.22 g), the actuator has a high power density of 1.7 kW/kg, an actuation strain of 300% under 80 g of external payload. We show how the actuator enables image depth control and an immersive tactile response in the form of augmented reality glasses and two-way communication haptic gloves whose thin form factor and high power density can hardly be achieved by conventional actuators.

[1] Department of Mechanical Engineering, Ajou University, 206 Worldcup-ro, Yeongtong-gu, Suwon-si, Gyeonggi-do 16499, Republic of Korea. [2] Samsung Advanced Institute of Technology, Samsung Electronics, 130 Samsung-ro, Yeongtong-gu, Suwon-si, Gyeonggi-do 16678, Republic of Korea. [3] Samsung Electronics, 34, Seongchon-gil, Seocho-gu, Seoul 06765, Republic of Korea. [4] Samsung SDS, 125, Olympic-ro, 35-gil, Songpa-gu, Seoul 05510, Republic of Korea. [5] Department of Electrical and Computer Engineering, Seoul National University, 1, Gwanak-ro, Gwanak-gu, Seoul 08826, Republic of Korea. [6]These authors contributed equally: Dongjin Kim, Baekgyeom Kim, Bongsu Shin. ✉email: sy84han@ajou.ac.kr; dskang@ajou.ac.kr; lhs12100@snu.ac.kr; jskoh@ajou.ac.kr

**W**earable devices have been developed for a variety of uses, including gaming, medical and communication applications[1,2]. In particular, wearable optical devices, such as augmented reality (AR)/virtual reality (VR) near-eye displays that enable an immersive visual experience, have attracted enormous attention in recent years[3–5]. For these devices, it is necessary to impart additional functions with actuators beyond simply displaying images[5–9]. For instance, a wide field of view, lightweight, and relieving visual fatigue should be considered simultaneously[6,7] for such devices to attract consumer's attention in the market. In addition, for vivid tactile sensation, there is a growing interest in wearable haptic gloves designed for high wearability, lightweight, compliance, and small form factor[10,11]. As a consequence, efforts to improve the optical performance of AR/VR near-eye displays[12,13] and compact haptic wearable devices[14,15] have included various actuation systems to implement additional functions, such as multi-focus AR glasses and thin form-factor haptic gloves (Fig. 1a, Supplementary Movie 1 and 2). AR glasses, one of the most widespread AR systems, have not been achieving remarkable levels of success in the market due to inconvenient wearability caused by large form factors and visual fatigue caused by vergence-accommodation conflict (VAC)[3,16]. The near-eye display using a focus-tunable lens intended to reduce VAC exhibits poor form factors and restricted platform flexibility[6]. The other way to relieve the VAC of AR glasses is mechanically adjusting the display with actuators allowing natural focus cues for users (Fig. 1b). In addition to wearable optical devices, non-vibrating mechanotactile outputs are important to generate natural and expressive tactile sensations on the skin through haptic devices[17]. To convey the sensation of a large skin deformation, haptic devices require actuators with a high force-to-weight ratio and a large displacement. Combining multiple actuators in the limited area of the haptic device also enables more expressive tactile experiences[18]. Actuators for such wearable devices are subject to a variety of design constraints, such as compact size, lightweight, and high power density. Under the volume and weight constraints for wearable devices, conventional electric motors on a small scale exhibit relatively low power density (Fig. 1c), which necessitates a prohibitively high actuator weight to achieve the desired output power.

Two inversely proportional features, a small form factor and a high payload, should be simultaneously satisfied for compact wearable actuating devices. Smart materials such as piezoelectric transducers (PZTs)[19–21], shape memory alloys (SMAs)[22–24], and dielectric elastomer actuators (DEAs)[25–28] are likely candidates (Fig. 1c). Piezoelectric transducers have advantages in their high power, energy density, and low power consumption, but they have an intrinsically low strain range and fragile mechanical properties. Bulky additional parts need to be combined to amplify stroke and attain robustness, leading to increments in size and weight[29]. DEAs not only have high power and energy density but also an excellent strain and relative contraction speed characteristics[29]. However, the fabrication of DEAs is complicated, and a DC-DC converter is necessary for high excitation voltage, adding to the cost and size of the device[30]. SMA actuators, which are used as artificial muscles for smart structures and soft robotics, have been studied for half a century[30–33]. However, despite their superior energy, power density[29,33,34], and ease of use, they have not made an industrial impact. SMAs have low energy efficiency[30,31] caused by thermal activation, small contraction, and nonlinear mechanical properties[35,36] arising from hysteresis in cyclic actuation and stress-induced phase transformation. In practical usage, most SMA actuators[37–40] have additional components that weigh much more than the bare SMA. This causes a decrease in the power and energy density by an order of magnitude, as shown in the schematic diagram in Fig. 1d.

Such limitations prevent SMA actuators from replacing commercially available DC motors and voice coil motors (VCMs) for compact devices such as wearable AR systems.

In this paper, by capitalizing on the principle advantages of the SMA actuator and adding a minimal strain-amplification mechanism to compensate for the limitations discussed above, we develop a simple but powerful actuator, the so-called compliant amplified SMA actuator (CASA) (Supplementary Movie 3). Previous studies[21,41] exploited the mechanism used to amplify the small strain of the SMA and the piezoelectric actuator. In contrast, we established an optimizing design to achieve relatively high actuation strain (56.1%), power density (1.7 kW/kg), and thin form factor (5 mm of height) with the desirable force and frequency for practical use in AR wearable applications. The CASA also combines an SMA wire and a compliant structure that amplifies the strain without heavy rigid components required for amplification mechanism in the previous study. Its simple compliant structure minimizes the reduction in the power and energy density of the raw material to achieve high actuator performance (Fig. 1d). Due to the simple and precise fabrication process, it can be easily designed at multiple scales. The actuator weighs 0.22 g but has a maximum actuation strain of 300% under 80 g of external payload, a payload 800 times higher than its weight, and a high power density of 1.7 kW/kg (>0.2 kW/kg for biological muscle). To surpass a CASA's peak force of 2N or stroke of 3 mm with a commercialized VCM, which is broadly employed in autofocus modules, the typical VCM actuator must be at least 70 times (>15 g) heavier or twice thicker (>12 mm) than the CASA.

Considering the strain induced by the SMA phase transition and the deformation of the elliptic configuration of the CASA, we establish an analytical model (Supplementary Fig. 1 and Table 1) to design an actuator with a maximum driving distance within a limited volume. We also integrate the actuator with a bistable parallelogram linear stage (BPS), which is a single-sheet compliant structure to compensate for the inherent limitations of SMAs and minimize undesired motions such as tilting and rotation by external forces. Bistable switching of the BPS eliminates the need for an input holding energy to maintain a certain stage state, which improves energy efficiency. As a practical application, we introduce a prototype pair of AR glasses and a soft glove-type haptic device. These glasses are capable of image depth control to relieve visual fatigue (Supplementary Movie 1) and the haptic glove generates high pressure (Supplementary Movie 2) to provide large skin-deformation sensation for the wearer without bulky actuation components (Fig. 1e). Last, we utilized the CASA as a resistance-type force sensor to detect external contact. It provides both sensing and actuation function without embedding separate actuators and sensors in the haptic device.

## Results

**Principles of operation, design, and characteristics of the CASA.** Prior to application of the CASA into practical applications, we first investigate how its operation relies on the bending of elastic beams caused by the contraction of a SMA wire, as shown in Fig. 2a, Supplementary Fig. 1c and Movie 3. The actuator consists of two elastic beams coupled together at the ends and a SMA wire embedded as multiple rows between the beams (Supplementary Fig. 3). The elliptic configuration of the actuator, which resembles a bow, amplifies the low actuation strain of the SMA wire. The optimally designed strain amplification configuration of elastic beams enables a relatively high actuation strain and large force compared to other amplification mechanisms (Supplementary Fig. 4 and Table 2). Figure 2b shows a CASA expanding in the vertical direction by the bending

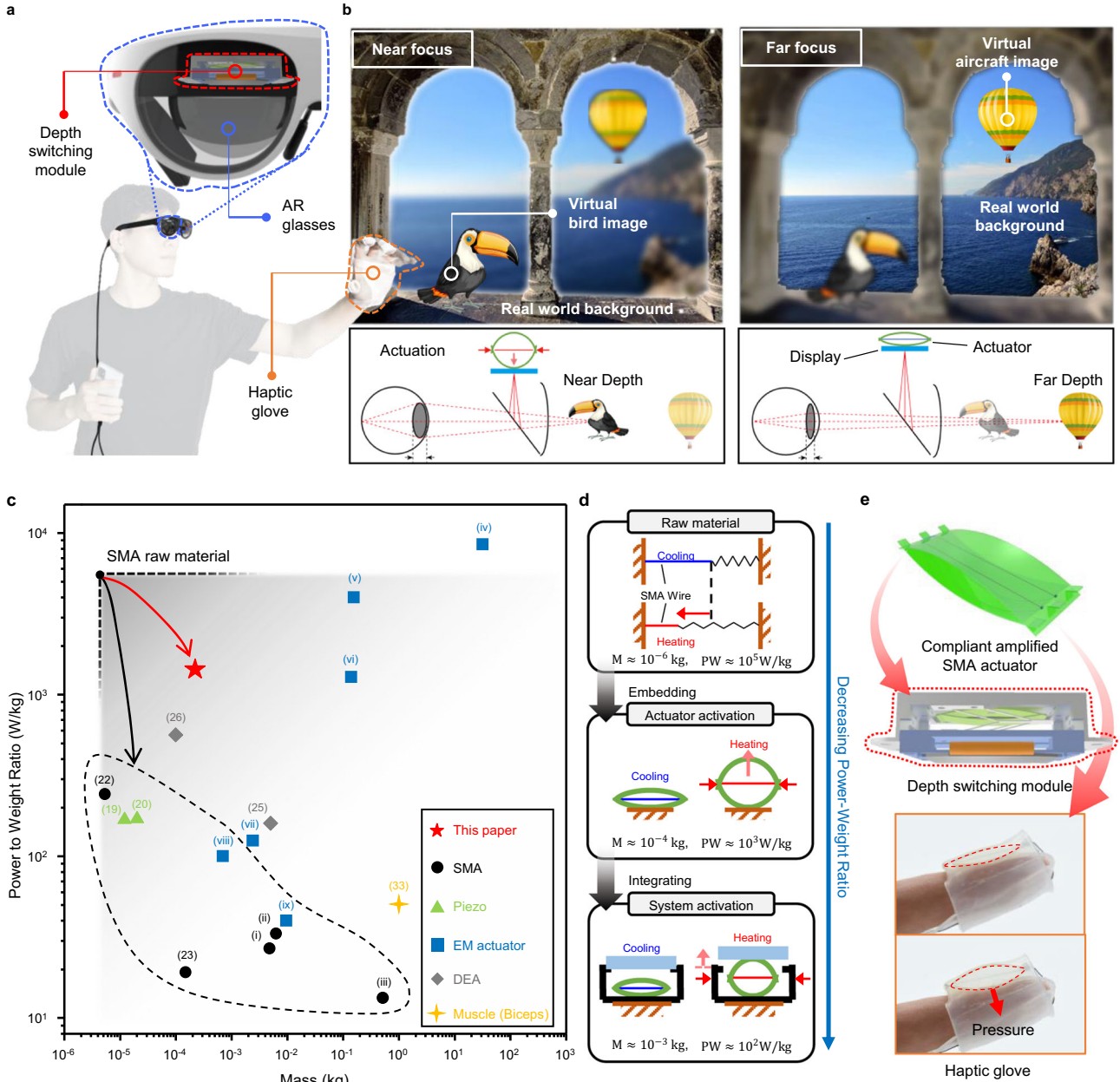

**Fig. 1 Overview of wearable devices using the CASA in an AR environment. a** User interaction with AR glasses featuring a depth-switching module and a haptic glove for an immersive experience. **b** Providing natural focus cues by switching the focusing depth of the image. Virtual images of a bird and an air balloon are labeled indicating near and far focus, respectively. The inset shows how the actuator is used to manipulate the display position with respect to the eye and create a natural focus on the different objects. **c** The power-to-weight ratio versus the weight of various artificial muscles, commercial actuators, and biological muscles. The red and black arrows represent the degradation of the power-to-weight ratio of CASA and existing SMA actuators, respectively. Data for Miga S125 (i), iNITIator-062 (ii), KLA05 (iii), Tesla Model S motor (iv), Scorpion (v), PTFF-K20 (vi), 22ECS60 (vii), MK04-24 (viii), and 828600 (ix) were taken from datasheets of the Miga Motor Company (i, ii), United States, Kinitics Automation Limited (iii), Canada, Tesla publications (iv), United States, Scorpion Power System Limited (v), United Kingdom, Pelonis Technologies. Inc. (vi), United States, Portescap (vii), United States, DIDEL SA (viii), Switzerland, and Crouzet Automatismes SAS (ix), France, respectively. **d** Schematic illustration of the raw material, embedded actuator activation, and integrated system activation as integrating components. The composite mass, M, and power-to-weight ratio, PW, are labeled for each level within the schematic. **e** Soft haptic glove for pressure sensation and direct display switching module for resolving VAC, both enabled by the CASA (green schematic).

motion of the beams upon contraction of the SMA wire. Our optimally designed actuator has an elliptic configuration and amplifies these small contractions of the SMA wire (typically 3–5%) to nearly 60% transverse actuation strain, as shown in Supplementary Fig. 5a. The initial volume can be substantially minimized by applying a preload, showing a 300.6% actuation strain (4.57 mm of actuation stroke and 1.376 mm of initial

height) under an 80 g preload when $L_b/L_s$ is 1.046 (Supplementary Fig. 5b), where $L_b$ is the length of the beam and $L_s$ is the length of the SMA wire (Supplementary Fig. 1a).

In addition to the actuation strain of the optimally designed CASA, actuation performances which are frequency, force, and strain can be achieved by designing three-dimensional parameters of the elastic beam and SMA wire, as shown in Fig. 2c. The

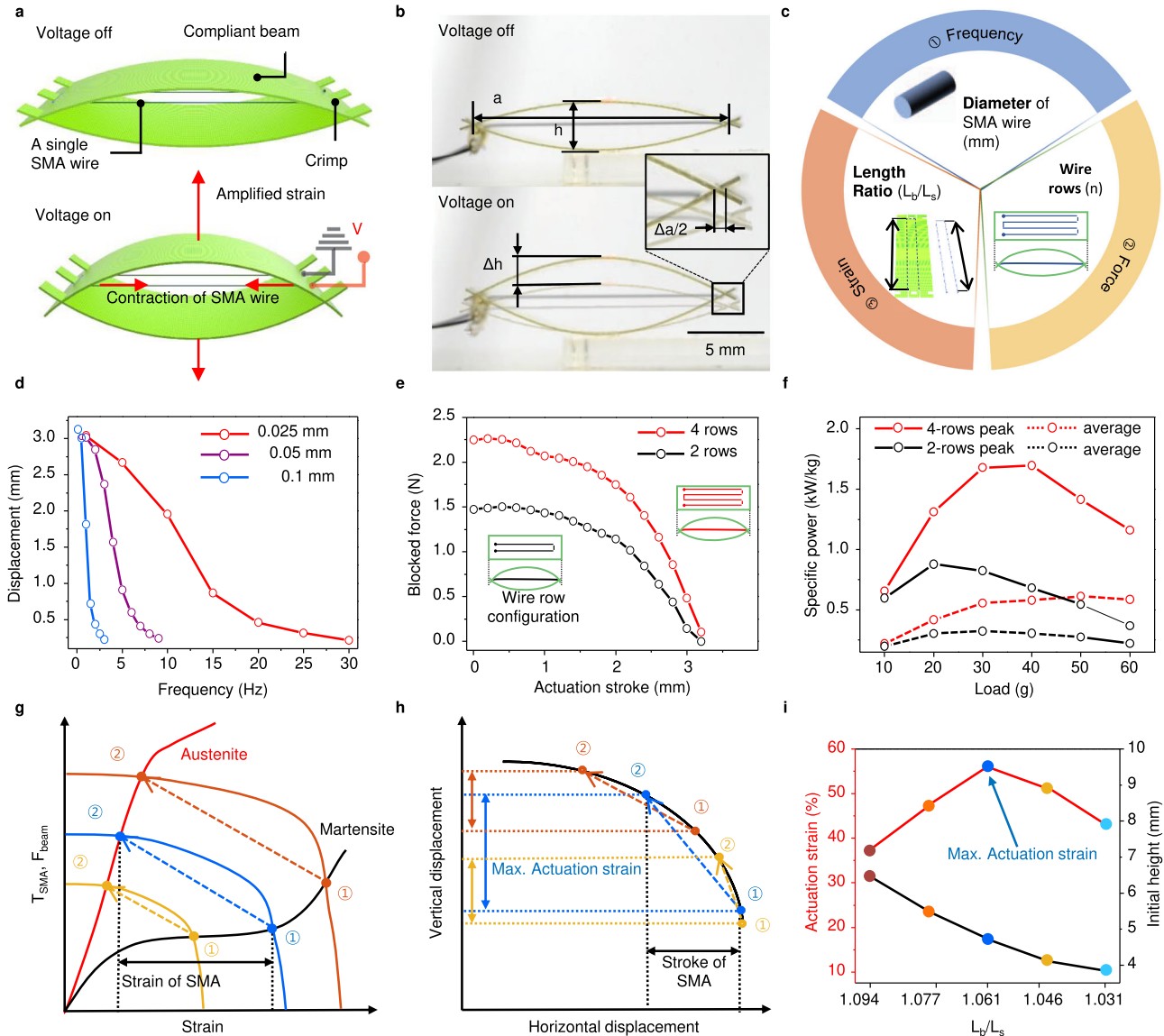

**Fig. 2 Principles of operation, optimal design of the CASA, and its performance. a** Schematic showing the actuation of the CASA, consisting of a pair of compliant beams and a single SMA wire arranged in parallel, by an applied voltage, *V*. The red arrows represent contraction of SMA wire and amplified actuation by compliant beams. **b** Optical images showing the linear behavior of the CASA upon actuation. The CASA height, *h*, and length, *a*. **c** Design process to achieve three actuation performances with three design parameters of CASA. **d** Actuation displacement as a function of frequency for SMA wire with a diameter of 0.025, 0.05, and 0.1 mm. **e** Blocked force as a function of actuation stroke for two and four rows of embedded SMA wire. **f** Peak and average specific power for two and four rows of embedded SMA wire as a function of preload. **g** Schematic plot of force–strain curves of the SMA and compliant beam. The black and red curves are force–strain curves of the martensite and austenite phases, respectively, of the SMA wire. The yellow, blue, and orange curves are force–strain curves of complaint beams buckled by different initial lengths of the embedded SMA wire (orange < blue < yellow). The comparison of the CASA actuation stroke with different initial SMA wire lengths under different preload weights. **h** Vertical displacement as a function of the horizontal CASA displacement. **i** Actuation strain and initial height as a function of the dimensionless length, $L_b/L_s$, where $L_b$ and $L_s$ are the length of the beam and SMA wire, respectively.

desired actuation frequency, force, and strain depend on the diameter of the SMA wire, rows of the embedded SMA wire, and the length ratio between the elastic beam and SMA wire, respectively.

To achieve the desired actuation frequency, we observed the frequency and displacement for three different diameters of the SMA wire, as shown in Fig. 2d. Selecting the diameter of the SMA wire is significant for improving the frequency because the transition time between the martensite and austenite phases is determined by the diameter of the SMA wire[42,43]. The time required for the transition from the martensite to austenite phase of the SMA wire can be greatly reduced by applying a high

voltage in a short time (~20 ms). Conversely, it is difficult to reduce the cooling time (from austenite to martensite) with no external cooling methods such as heat sink materials, forced air, or heat conductive grease, which require additional components or complex fabrication processes. To reduce the cooling time of SMA wire for a wide range of actuation frequency, we embedded SMA wires with the diameter of 0.025, 0.05, and 0.1 mm (Supplementary Table 3) and observed that the actuator can actuate under up to a high frequency of 30 Hz and a displacement of 0.2 mm as shown in Supplementary Fig. 6.

However, embedding a thinner SMA wire for a high-frequency results in a reduction of actuation force. This can be compensated

by multiple rows routed with a single SMA wire, as shown in Fig. 2e and Supplementary Fig. 3d. The addition of multiple wire rows in the actuator distributes the tension applied to the entire SMA wire. This enables the CASA to endure more preload and lift heavier objects than with fewer rows of embedded SMA wire. Figure 2e compares the blocked force of actuators with two and four rows of embedded SMA wires, showing that adding more wire rows can efficiently scale up the force while maintaining the actuation strain. To measure the maximum specific power (power-to-weight ratio), we applied a high voltage for 20 ms to examine its actuation speed and acceleration under different inertial loads. As shown in Fig. 2f, we measured a maximum value of 1696 W/kg during actuation with a weight of 40 g, and the average specific power was 651 W/kg (Supplementary Movie 4 and Supplementary Fig. 7). The maximum value of the specific power of the CASA is higher than that of another artificial muscle actuator (Fig. 1c).

Defining the optimal state of equilibrium for the initial prestrain of the SMA wire and the initial buckling of elastic beams are key experimental steps toward achieving a high actuation strain. Figure 2g shows how varying the initial length of the SMA wire controls the equilibrium states of the SMA wire and buckling beams, thus determining the strain of the SMA wire. The red and black SMA wire tension–strain curves meet the orange, blue, and yellow buckling beam force–strain curves at the intersection points where the equilibrium force is applied, denoted as circles numbered 1 and 2, respectively. Using a relatively short length of SMA wire embedded in elastic beams represented as an orange curve induces these beams to buckle more than for a longer SMA wire, resulting in a higher equilibrium force. Upon activation under this high tension, the wire deforms through a full transition from martensite to austenite (from circle numbered 1 to circle numbered 2 of the orange curve). The strain of the SMA wire is maximized under high tension, as represented by the orange curve in Fig. 2g.

However, maximizing the actuation strain of the SMA wire does not directly result in maximum transverse actuation strain of the actuator. Figure 2h demonstrates that the incremental ratio of the vertical-to-horizontal displacement decreases as the initial height ($h$) increases. As a result, the actuation stroke (vertical displacement) of the blue dashed line is larger than that of the orange dashed line even though the actuation stroke of the SMA wire (horizontal displacement) of the orange dashed line is larger than that of the blue dashed line, as shown in Fig. 2h. Therefore, to optimize the actuation strain, it is important to find the optimal length of the SMA wire to attain a sufficiently high prestrain of the SMA wire while minimizing the initial height of the elastic beams (Supplementary Fig. 8). The numerical results of these optimized actuation strain and displacement for different lengths of SMA wire are described in Fig. 2i. We demonstrated that the optimal design amplifies a 3.3% actuation strain of the SMA wire in the horizontal direction to 56.1% actuation strain of the CASA in the vertical direction. The sample length, actuation time, applied voltage, and current of the SMA wire in each test are presented in detail in Supplementary Table 6.

Moreover, the actuation stroke and force can be scaled up by serially connected actuators (Supplementary Movie 5) and embedding multiple arrays of SMA wire (Supplementary Movie 6) as shown in Supplementary Fig. 10 and Supplementary Note 1. The CASA with the multiple arrays of SMA wire lifts up 300 g of weight which is about 800 times heavier than its own weight (Supplementary Fig. 10e).

**Bistable parallelogram linear stage to improve energy efficiency.** To implement SMA actuators into wearable devices with

limited battery capacity, power consumption should also be taken into account. The CASA requires continuous input energy to maintain a given configuration or state and, even then, unpredictable disturbances affecting the actuator can degrade the position accuracy. To compensate for these limitations and minimize the overall power consumption, we combine the BPS with the actuator.

The BPS is a single-sheet compliant mechanism that has buckling springs, inducing the bistability of the stage to increase the position accuracy and energy efficiency (Fig. 3a and Supplementary Movie 7). The stage achieves motion stability through a parallelogram mechanism that constrains the tilting and rotation of the moving platform (Fig. 3b). Its single-sheet structure also minimizes the volume and mass of the linear stage. Therefore, the BPS is suitable for application to small-scale devices such as actuators for depth-switching modules in AR glasses as shown in Supplementary Fig. 11.

In addition, the bistable stage compensates for the inherent limitations and disadvantages of SMAs, which have low controllability[30] and low energy efficiency. The low controllability and inaccurate positioning of the SMA actuator are caused by nonlinearity in the martensitic transformation and thermal hysteresis of SMA materials[30]. The low energy efficiency of the SMA is caused by high thermal losses in the actuation process (e.g., latent heat for phase transition and convection). To overcome these limitations, we develop a simple SMA actuator module (Supplementary Fig. 11a) combining CASA and BPS considering three objectives.

First, we use the BPS to enhance the energy efficiency of the CASA with zero holding current by bistability, which means that input energy is only required to shift the stage position state (Fig. 3c and Supplementary Note 2). Because the actuator with a bistable stage maintains an actuation stroke without an applied current (Supplementary Fig. 12b and c), this enhancement reduces the energy consumption compared to the actuator without a bistable stage (Supplementary Fig. 12d). Second, the bistable structure induced by the energy barrier of the buckling beam improves the previously low positioning controllability of the SMA (Fig. 3d and e). The bistable structures maintain their states at low energy levels, as shown in Fig. 3e. Thus, the bistable stage maintains its accurate position without an additional actuation force. Applying a varying actuation frequency confirms that the actuator with a BPS achieves a higher position accuracy than the actuator without a BPS (Fig. 3f). In addition, the snap-through effect in the transition between the two stable states generates much higher output work than the actuator alone at higher actuation frequencies (Fig. 3g). Third, the compliant parallelogram mechanism constrains undesired degrees of freedoms, such as tilting and rotation, while still enabling parallel linear motion in the vertical direction. The bottom images in Fig. 3b show that the bistable stage without a parallelogram cannot sustain tilting and rotating upon off-axis external loading. In optical systems, a small amount of tilting and rotation of the display causes large image distortion since the optical components in AR glasses usually magnify the image. By applying a BPS to actuate the display panel, the compliant parallelogram mechanism improves motion linearity despite external perturbations.

The modeling for force-displacement relationship of the BPS (Supplementary Fig. 1d and Table 4) and its experimental results are shown in Fig. 3d. From these data, the potential energy can be calculated, as shown in Fig. 3e, and both the reaction force and energy barrier $\Delta E$ of the bistable system can be obtained. This energy barrier is the required threshold energy value for transforming the state of the stage. The bistable stage shifts its position upon the application of a force that exceeds the energy barrier, as shown in Fig. 3d and e. The CASA with a BPS can

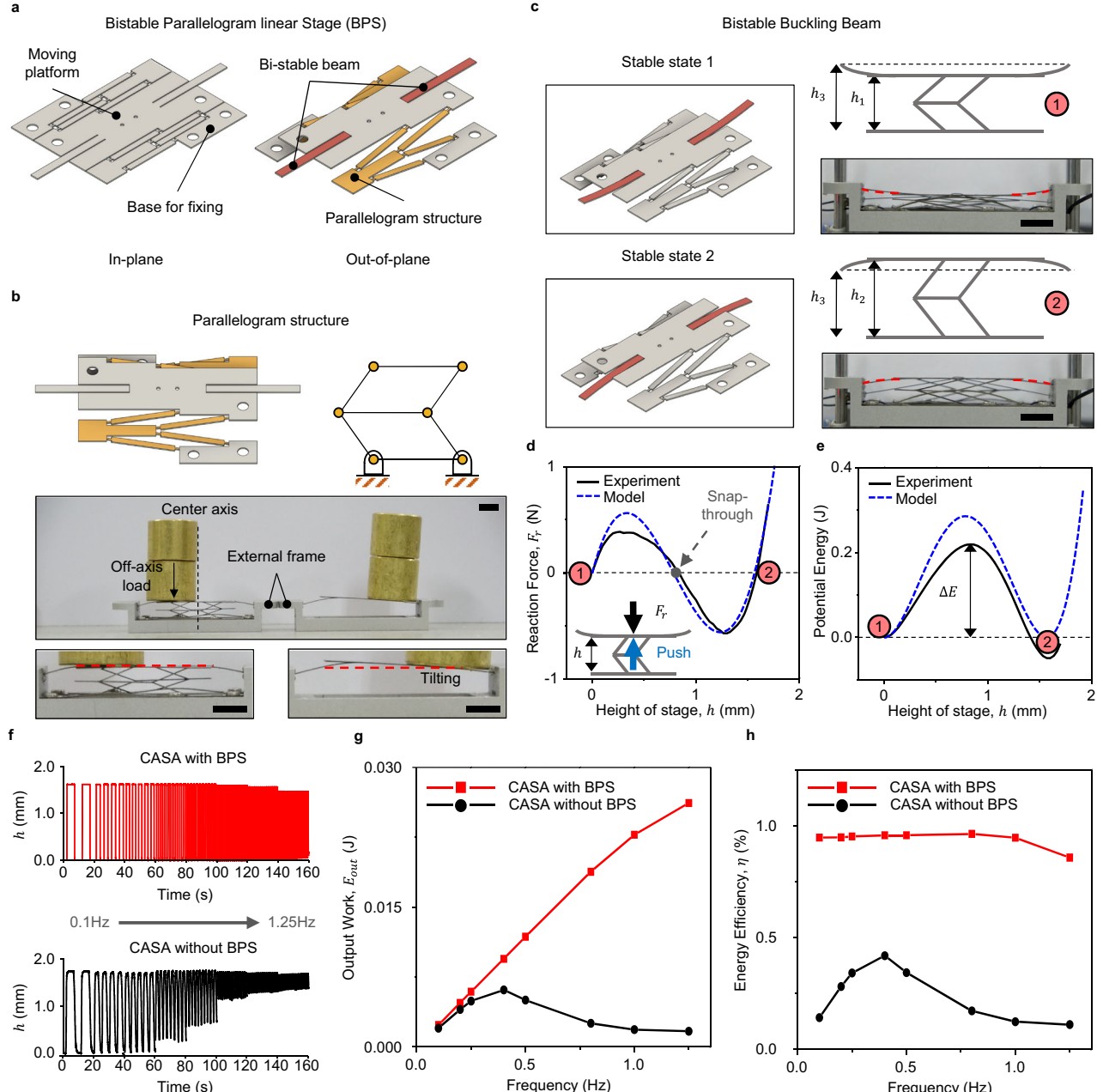

**Fig. 3 Low-profile BPS for improving the CASA energy efficiency and position accuracy. a** Schematics of (left) in-plane BPS components and (right) neutral BPS position before engaging the buckling beam on both sides for bistability. **b** (top-left) Illustration of the compliant parallelogram linkage for preventing stage tilting. (side view) Schematics of the conventional parallel 4-bar mechanism corresponding to the compliant parallelogram mechanism. (bottom) Comparison of the tilting of the bistable stage with and without the compliant parallelogram mechanism. **c** (top/bottom left) Two stable BPS states. (top/bottom right) Side view of schematic photographs for the two stable states, labeled using red circles. Stable states 1 and 2 correspond to the height $h_1$ and $h_2$, respectively. The height of $h_3$ is the neutral position without beam buckling. **d** Vertical reaction force profile of the BPS with the height of the stage, $h$. The two stable states 1 and 2 (red circles) and the unstable state inducing snap-through (gray dot) are labeled. **e** Corresponding potential energy curve describing the experimental (black line) and simulated (blue dashed line) results. **f** Actuation stroke of the CASA with BPS (upper plot) and CASA without BPS (lower plot) with increasing actuation frequency. The frequency was increased every 20 s in gradual increments, starting from 0.1 Hz. **g** Comparison of the output work ($E_{out}$) and **h** energy efficiency ($\eta$) for the CASA with BPS (red) and CASA only (black) with varying actuation frequency from 0.1 to 1.25 Hz for 20 s.

maintain its stroke at both low and high actuation frequencies as shown in Supplementary Fig. 15. In addition, maintaining the stroke at a high frequency enables high output work (Fig. 3g). Due to this high output work, the actuator with a BPS can achieve an energy efficiency of ~10 times higher than the actuator without a BPS (Fig. 3h). This efficiency of ~1% is quite close to that of the raw SMA actuator.

**Actuating compact AR devices**. The proposed actuators are utilized to perform direct image-depth switching in the AR glasses prototype (Fig. 4a). This prototype is tethered to and powered by a cellphone. The AR glasses prototype exhibits a clear image displaying a wide 60° field of view using a 0.7″ FHD micro-OLED (Organic light-emitting diode) display and air birdbath architecture which consist of half mirror and curved mirror

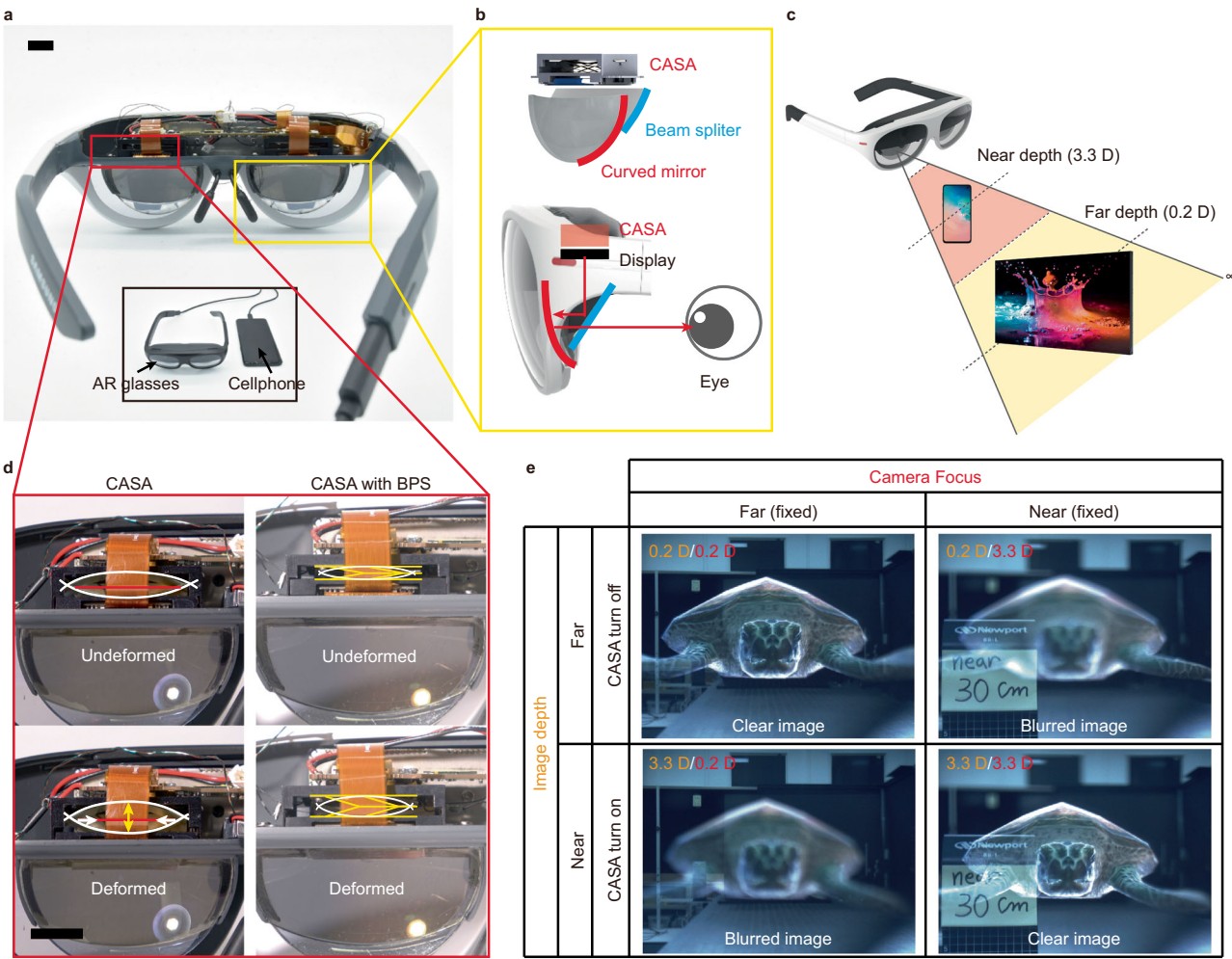

**Fig. 4 Application of the CASA and BPS to prototype AR glasses. a** Multifocal air birdbath type AR glasses prototype with the combined CASA/BPS device. (inset) AR glasses prototype is tethered to and powered by a cellphone. **b** (top) The architecture of air birdbath type combiner. (bottom) The half mirror reflects the image from the display to a curved mirror, which magnifies the image and redirects it to the eye. **c** The image depth selection according to the use case of 2D contents. **d** Comparison of CASA and CASA with BPS which are adopted to AR glasses prototype. **e** Comparison of clarity of a virtual image according to actuation of CASA. Depth of focus of the camera (column) and depth of virtual image (row).

(Fig. 4b). The thickness of the air birdbath architecture is 15 mm. Even including the sun visor, the total thickness of the combiner is only 20 mm, which is similar to that of general waveguide type AR glasses[9]. Using a cellphone powered by a 3000 mAh battery, the CASA with a BPS can actuate for ~$3.2 \times 10^5$ cycles.

For 2D contents, appropriate image depths are manually selected (Fig. 4c). For displaying 3D objects, the position of the display is adjusted by the CASA according to the relative distance between the user and the 3D objects. Figure 4d shows the implementation of the actuator with and without a BPS to an AR glasses prototype (Supplementary Movie 8). Previously, we have discussed an optimal design that can implement the maximum actuation stroke within a limited form factor. In the case of the proposed AR glasses prototype, an actuation stroke of ~1 mm is required to adjust the position of the virtual object from 0.2 D to 3.3 D, where D is the diopter which is equal to m$^{-1}$. Therefore, the CASA and BPS are designed for 1 mm of actuation stroke to achieve a switchable image depth, which cannot be readily achieved by conventional actuators with similar form factors and weights. The required stroke for the target depth depends on the corresponding optical power of the combiner; here, 0.2 D and 3.3 D. The actuator in the undeformed state provides far-depth (0.2 D) images. As the actuator deforms, the image smoothly switches

from far to near depth (3.3 D) within 300 ms (Supplementary Fig. 16 and Movie 1). The CASA can restore to its original state by the compliant beams. However, the BPS requires an additional actuator to restore to its original state (from 0.2 D to 3.3 D) after the actuation. We employed the counter SMA wire passing through holes located in the center of the bistable stage and anchored at the external frame as shown in Supplementary Fig. 11a. To return the stage, the counter SMA wire contracts by applying an electric current and pulls the moving platform of the BPS. When the actuation force of the counter SMA wire exceeds the BPS energy barrier, the bistable stage returns (Supplementary Fig. 11f).

The robustness of the proposed actuators ensures suppression of tilting and the precise switching of the image. Figure 4e shows that the position of the image is accurately adjusted from far to near depth and vice versa. Based on the calculated distance range where the 3D contents should be located, the CASA responses to shift the depth of the 3D image. As a consequence, we applied the actuator with and without a bistable stage to build AR glasses prototype with a significantly small form factor (0.2 g and 4-mm thickness) that show remarkably superior image quality compared to that of previously proposed varifocal AR near-eye displays[5,8].

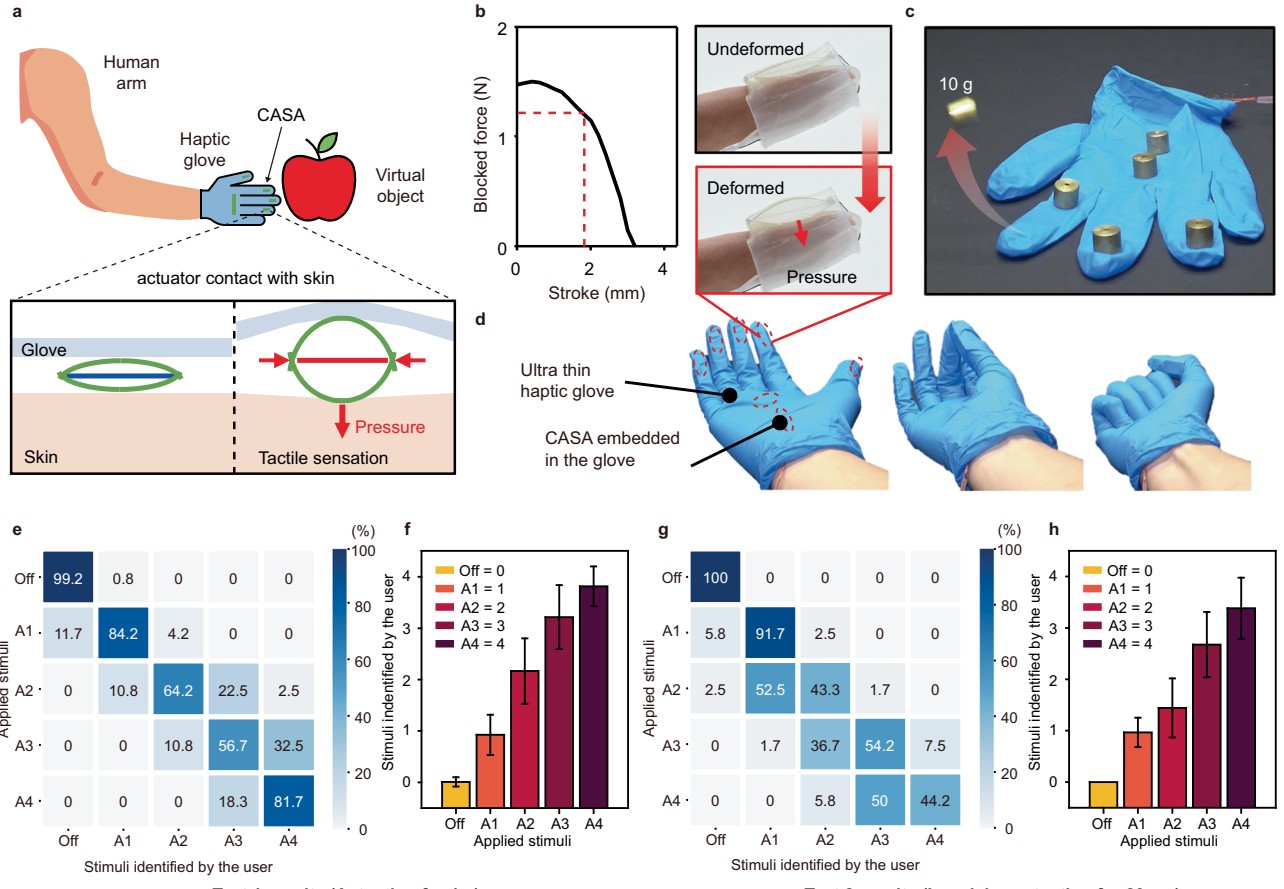

**Fig. 5 Actuation of the CASA in the haptic glove. a** (top) Conceptual illustration of the interaction between a human and a virtual image in AR using haptic gloves with the CASA. (bottom) Actuation mechanism of the CASA in a haptic glove for realizing tactile sensation. **b** (left) Force-displacement relationship of the CASA in the fingertip of the haptic glove. (right) Pressure transmission mechanism visualized with the transparent haptic glove integrated with the CASA on the fingertip. **c** High power actuation of thin haptic glove integrated with CASA throwing 10 g of the weight. **d** Thin haptic glove integrated with multiple CASAs minimizing interference during various motions of fingers. The red dashed circles indicate CASAs embedded in the haptic glove. **e** The confusion matrix of Test 1, showing identification rate (IR) of 12 untrained volunteers. **f** Average reported feeling of users (scale 0 to 4) versus Applied stimuli in Test 1. **g** The confusion matrix of Test 2, showing Identification Rate (IR) of 12 untrained volunteers. **h** Average reported feeling of users (scale 0 to 4) versus Applied stimuli in Test 2. Error bars show mean ± standard deviation.

Furthermore, the high power density and scalability characteristics of the CASA allow its utilization to actuate haptic gloves, as shown in Fig. 5. In addition to providing a visually immersive experience by AR glasses, a haptic glove equipped with CASAs also provides a high-pressure tactile experience with a relatively high continuous contact pressure compared to high-frequency actuators (vibration) under a limited volume (Supplementary Movie 9). The high power of the actuator enables thin and soft haptic gloves to generate ~12 kPa of pressure (Fig. 4g) without bulky and rigid actuators that are used in relatively heavy haptic devices[10]. Unlike actuators able to provide contact stimuli through vibration, the CASA provides a high force and actuation stroke to convey the sensation of large static skin deformation. Using a transparent silicone rubber, the actuation in the haptic glove is visualized. We integrated seven number of actuators in a thin polymer glove (Fig. 4h). Despite its thin form factor, the haptic glove is so powerful as to throw a weight of 10 g. The haptic glove can, thus, convey the highly impulsive pressure as well as gradually increasing and static pressure to the user. The simplicity, thin form factor and compliance of the haptic glove allow for a naturally fit condition without impairing any action of the fingers (Fig. 4i). These favorable properties of our haptic glove

indicate that the choice of the actuator for the wearable system is central, which is typically the heaviest and bulkiest part.

For quantitative tests, we recruited 12 volunteers (3 females and 9 males, ranging in age from 25 to 32 years old). They wore the haptic glove on the left hand. The CASA on the index fingertip was only actuated to convey the sensation as shown in Supplementary Fig. 17. Two types of tests were performed to measure the signal intensity perceived by the users for different magnitudes of the contact forces. The user perceived gradually increasing contact forces generated for 1 s in Test 1 and impulsive forces generated for 20 ms in Test 2 as shown in Supplementary Fig. 17. The generated peak forces of 4 different magnitudes were ranged from 0.083 to 2.052 N in Test 1 and from 0.045 to 1.135 N in Test 2. Before the test started, each user was shown all stimuli once and told the corresponding stimulus number for each (A1, A2, A3, and A4).

Figure 5e shows the confusion matrix, in which the rows and columns corresponds to the applied signal and signal identified by the user in Test 1, respectively. The users have a high Identification Rate (IR) of over 80% for the lowest and highest magnitudes of forces (A1 and A4). For A2 and A3, the users have an IR of over 55%. After experiencing many series of stimuli, the

users are unsure about identifying the specific signal, especially the two middle stimuli (A2 and A3). However, the averaged intensity of each signal in Fig. 5f and Supplementary Fig. 18 show that the users are able to compare the different intensity during the test in overall.

Figures 5g and h and Supplementary Fig. 19 show corresponding perception results of Test 2 measuring how users perceive impulsive forces in an instant (20 ms). The users correctly distinguish the off state and the lowest magnitude of sudden force with a high IR of over 90%. As shown in Supplementary Fig. 17b the differences of peak force in Test 2 are comparatively low. The differences were ranged from ~0.13 to 0.71 N. Due to the relatively low difference and actuation time, the users have an IR between 40 and 55%. Similar to Test 1, the averaged intensity in Fig. 5h shows the users are able to compare the intensity of the signal overall.

The perception results demonstrate that the CASA is able to convey various sensation, such as large skin deformation (2 N) and gentle touch (0.05 N), at different actuation time.

**Sensing capability for tactile communication**. In addition to actuation, the sensing function is highly desirable in wearable systems[44,45]. The tight integration of the actuation and sensing[46–48] provides multifunctional devices increasing application potential for next-generation wearable systems. The CASA with the embedded smart material embedded has a favorable characteristic to couple sensing and actuation for applications such as communication for deaf-blind people through the haptic glove.

The electrical resistance of the SMA can be varied by changing the strain. The SMA-based device can be controlled through resistance-based self-sensing techniques[49,50]. Figure 6a shows that the CASA can be utilized as a resistance-type force sensor for measuring the external contact. Before pressure is applied, the compliant bending beams and routed SMA wire with constant resistance (R1) are in equilibrium. External pressure causes the bending compliant beams to straighten, resulting in the extension of the SMA wire. The resistance of the SMA wire, thus, increases (R2). Once the external pressure is removed, the compliant beams and the SMA wire are restored to their original states, and the resistance also returns to its original value (R1).

Figure 6b shows the loading and unloading characteristics of the two CASAs with different diameters of SMA wire (0.1 mm and 0.025 mm). The 0.025 mm diameter SMA wire and 0.1-mm-thick compliant beams exhibits a relatively high sensitivity of $0.1498\,N^{-1}$ under an applied force of 0.13 N. The 0.1 mm diameter SMA wire and 0.2-mm-thick compliant beam exhibits a sensitivity of $0.01595\,N^{-1}$ under an applied force of 1.1 N. The sensitivity and measurable force range are determined by the beam thickness and wire diameter. The high stiffness of the CASA with the thick beam (0.2 mm) and thick wire (0.1 mm) enables a wide range of measurable force, while the low stiffness of the CASA with the thin beam (0.1 mm) and thin wire (0.025 mm of diameter) provides high sensitivity. During unloading, the response characteristic aligns with the corresponding loading characteristic without noticeable hysteresis under the applied force. The results of the loading–unloading test conducted for ~1000 cycles demonstrate the durability of the CASA as a force sensor (Supplementary Fig. 20).

The CASA exhibiting sensing capability and high actuation performance can be used to develop highly compact haptic gloves for the tactile communication described in Fig. 6c without embedding separate actuators and sensors (Supplementary Movie 10). Tactile communication allows deaf-blind people to communicate through a Braille-based device[51–53]. A Braille cell consisting of six dots that can represent letters, numbers, and special characters with the combination of raised dots[53]. Through the finger Braille communication method, the user can represent the six dots of Braille using the index, middle, and ring fingers of both hands[52].

As shown in the scenario in Fig. 6c, users are wearing the haptic gloves with the CASAs embedded on the index (L3, R3), middle (L2, R2), and ring (L1, R1) fingers of both hands. User A presses the corresponding fingers on the ground to represent the Braille code as shown in Supplementary Fig. 21. The CASAs embedded on each finger are also applied by contact force and the letter represented by the Braille code can be identified based on the resistance changes in the SMA wire. The resistance data can be used to actuate the corresponding CASAs embedded in user B's haptic glove, allowing user B to receive the message. User B can send messages to user A in the same way.

Figure 6d and e shows the letters H, O, W, A, R, E, Y, O, U, ? and G, R, E, A, T, ! are typed by users by tapping the appropriate combination of the fingers on the desk with the haptic glove. The resistance changes in the CASAs as sensors are clearly identified for each letter. The previously demonstrated actuation characteristic of the CASA enables the user to sense received messages as well.

## Discussion

The compliant amplified shape memory alloy actuator (CASA) is distinguished by its high performance of actuation strain and power, with a lightweight and a thin form factor, demonstrating the practical use of smart materials in the commercial market of wearable devices. The fabrication scalability and comprehensive experimental studies of the actuator enable the employment of the CASA in a broad spectrum of potential applications, including consumer electronics, intelligent robots, and medical and communication technologies. However, the low precision of the strain control and high energy consumption rate are inherent limitations of SMA-based actuator. By combining the resistive sensing feedback, the strain control of the SMA can be improved, which is advantageous for developing the multifunctional actuator capable of force sensing without additional strain sensors, as the biological muscle does. In terms of energy consumption of SMA-based actuators, the absolute value of energy consumption in micro-scale applications is low enough for battery use. Using a mechanical design with passive appendages, such as the bistable structures used in this study, also increases the working time of SMA-based actuators. In addition to wearable device applications, future work will attempt to develop a platform for on-textile interfaces[54] for clothing actuation keeping wearer comfort in mind.

Our work shows that smart materials, specifically artificial muscle actuators, provide a smart solution to overcome the maximum power and energy density limitations of conventional actuators for application to compact wearable devices.

## Methods

**CASA and BPS fabrication procedure**. The linear actuator module (Supplementary Fig. 11a) of the optical wearable device (AR glasses) comprises an OLED display panel (5 g) (Supplementary Fig. 11b), a BPS (100 mg) (Supplementary Fig. 11c), a CASA (50 mg), an external frame, and a lower part. This linear actuator module is a key component required to lift the 5 g weight of the OLED display panel, despite the constrained form factor of the AR glasses.

The CASA comprises two compliant beams fabricated by laser machining a glass-fiber composite sheet (thickness of 100 or 200 μm) and a both-ends-clamped SMA wire (diameter of 0.025, 0.05, or 0.1 mm as shown in Supplementary Table 3). Compliant beams with a castellation pattern are fabricated by laser machining as shown in Supplementary Fig. 3a. This pattern allows each beam to be connected to form the bow shape of the CASA. The pattern also allows the SMA wire to be hung on it. After attaching crimps at the ends of the wire as shown in Supplementary Fig. 3b, the single line of SMA wire is anchored on the pattern of the compliant

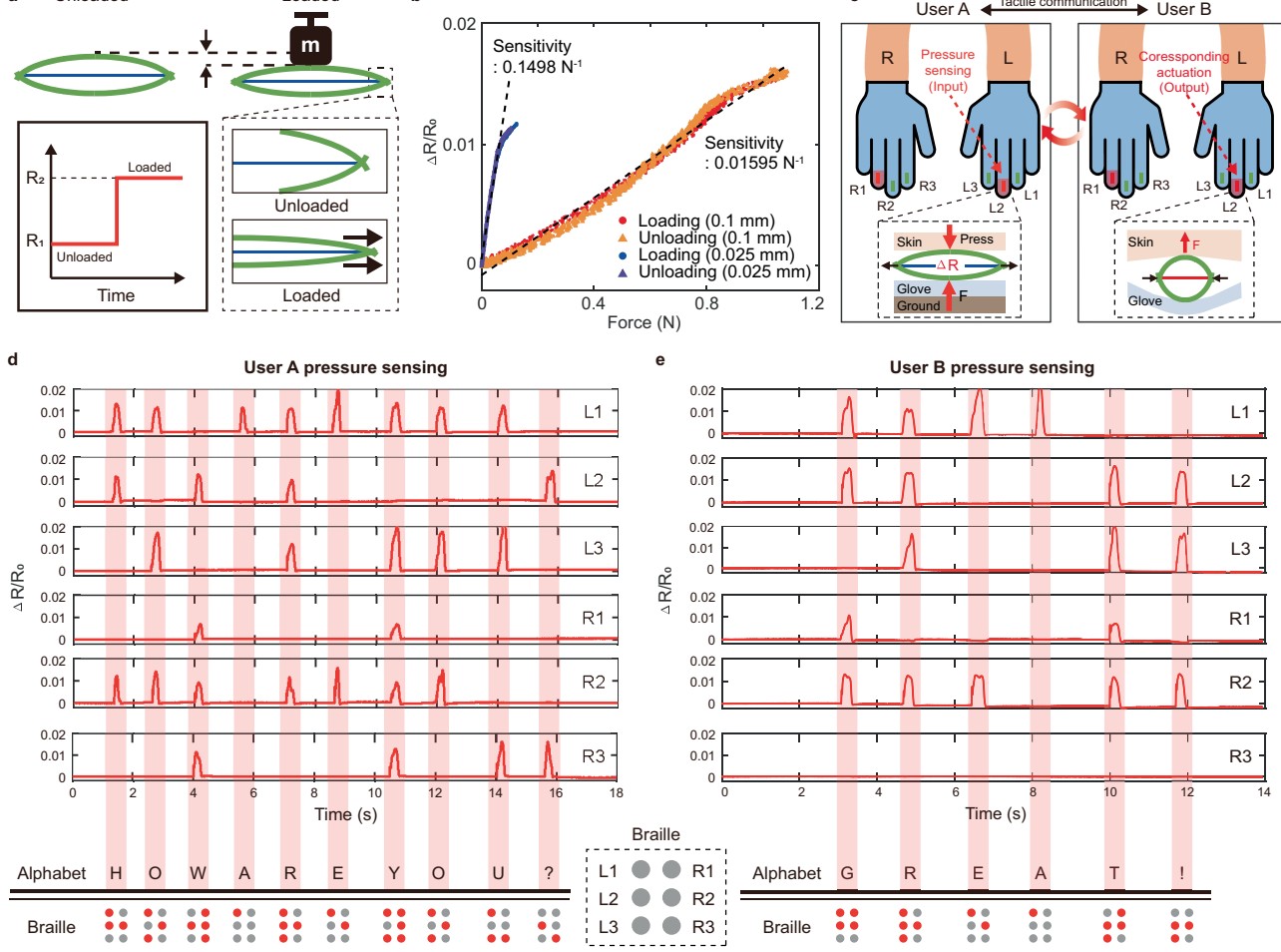

**Fig. 6 CASA with sensing capability and its application to tactile communication. a** Schematic of CASA's sensing capability measuring the external contact. **b** Normalized resistance as a function of applied force under loading–unloading test. **c** Concept of tactile communication using actuation and sensing capability of the CASA. **d, e** Normalized sensor output for each finger and corresponding Braille code to input H, O, W, A, R, E, Y, O, U, ? and G, R, E, A, T, !, respectively.

beam routed along the red dashed line and fixed at the small hole (red circle) at the ends of the crimps, as shown in Supplementary Fig. 3c. This embedded SMA wire induces the compliant beam to buckle. Especially, a single line of crimped SMA wire is embedded in the compliant for a desired number of rows as shown in Supplementary Fig. 3d. Subsequently, both buckled compliant beams are joined with the pattern (blue rectangles), as shown in Supplementary Fig. 3e. The castellation patterns (blue rectangles) in both compliant beams connect to each other and actuate as revolute joints. Consequently, the elliptical configuration of CASA as shown in Supplementary Fig. 3f amplifies the strain upon activation.

The BPS is a single sheet based on a compliant mechanism that can be employed as a linear stage (Supplementary Fig. 1b). The BPS is fabricated by laser machining a 150-μm stainless steel 304 sheet and has three components: a buckling beam for inducing bistability, compliant linkage, and living hinge as a joint with parallelogram mechanism, as shown in Fig. 3a and Supplementary Fig. 1b. The buckling beams assume the form of a rectangular shape whose width and length determine the load bearing and actuation strain of the stage. Compliant linkages and living hinges are components of the parallelogram mechanism. In addition, the width and length of the living hinge can determine the structural stiffness of the parallelogram. This structural stiffness allows for the stability of the linear motion of the BPS. A design based on compliant mechanism modeling is presented in the following section.

After fabricating BPS by laser-machining, the cut sheet is popped up to create a compliant parallelogram linkage, as shown in Supplementary Fig. 11d. The height of the popped-up BPS, $h_3$, corresponds to that of the pivot in the external frame. The buckling beams are contracted to fit into the pivots of the external frame, which results in the bistability of the BPS. The contraction length of the buckling beam is denoted as $l_c$ in Supplementary Fig. 11e.

Two actuators, the CASA and counter SMA wire, are used for the cyclic actuation of the BPS by shifting the state from 1 to 2 and vice versa (Supplementary Fig. 11f). The CASA embedded in the actuator module is composed of two compliant beams, as shown in Supplementary Fig. 10c. One beam has a connecting

hole, which is combined with a lower part in Supplementary Fig. 10a. The CASA combined with the lower part is bolted to the external frame. The counter SMA wire passes through holes located in the center of the BPS. Both sides of the counter SMA wire are clamped and anchored at the external frame. To actuate the CASA, voltage is applied to SMA wire embedded in the CASA. When the actuation force exceeds the energy barrier $\Delta E$ of the BPS, it shifts the state from 1 to 2. For shifting the state from 2 to 1 a counter SMA wire, clamped at the external frame as mechanical ground, is employed. By applying an electric current, the counter SMA wire contracts and pulls the moving platform of the BPS. When the actuation force of the counter SMA wire exceeds the BPS energy barrier, the state shifts from 2 to 1 (Supplementary Movie 7).

**Modeling of CASA and BPS.** The SMA wire and compliant beams, which are the main components of CASA, were designed to attain the desired actuation strain and power output under a limited volume of the actuator. Owing to the symmetrical configuration of the two buckled beams, a quarter of its shape, which deforms like a flexible cantilever beam, can be considered as a pseudo-rigid-body[55] (PRB) link, as shown in Supplementary Fig. 1c. The length and height of CASA, $a$ and $h$, respectively, are expressed by equations parameterized in terms of the PRB angle, $\theta$ :

$$a = L_b - L_b r \left(1 - \cos\theta\right) \quad (1)$$

$$h = L_b r \sin\theta \quad (2)$$

where $L_b$ and $r$ are the length of the beam and characteristic radius factor, respectively. The numerical value of the characteristic radius factor is determined by the angle of the force exerted on the beam. We measured the height ($h$) for different length of SMA wire ($L_S$) during actuation. Using Eqs. (1) and (2), the length ($a$) was calculated from the measured height ($h$), as described in Supplementary Fig. 8c. Actuation stroke is the change in the initial (before applying voltage) and final height (after applying voltage). The highest actuation stroke was measured for

$L_b/L_S = 1.061$, where the high incremental ratio of the vertical-to-horizontal displacement of CASA and sufficient actuation strain of the SMA wire are both satisfied.

The length of the CASA ($a$) also corresponds to the length of the embedded SMA wire, and the tension of the SMA wire equals the buckling force of the elastic beams, resulting in the equilibrium of force. The force ($F_{beam}$) exerted on the buckling beam and the tension of the SMA wire for austenite ($T_{aus}$) and martensite ($T_{mar}$) phases are represented as:

$$F_{beam} = \frac{1}{L_b^2 \sin\theta} K_\theta \theta E I \qquad (3)$$

$$T_{aus} = A E_{aus} \varepsilon \qquad (4)$$

$$T_{mar} = A E_{mar}(\varepsilon - \varepsilon_L \nu_{mar}) \qquad (5)$$

Here, $K_\theta$, $E$, $I$, and $A$ are the stiffness coefficient, flexural modulus of the beam, moment of inertia of the beam, and cross-sectional area of the SMA wire, respectively; $E_{aus}$ and $E_{mar}$ are the Young's modulus of the SMA wire for the austenite and martensite phases, respectively; $\varepsilon$ and $\varepsilon_L$, and $\nu_{mar}$ are the strain of the SMA wire, residual stress, and martensite volume fraction, respectively. The martensite volume fraction is:

$$\nu_{mar} = \frac{1}{2} \cos\left(\frac{\varepsilon^{cr} - \varepsilon}{\varepsilon^{cr} - 0.001}\right)\pi + \frac{1}{2} \qquad (6)$$

where $\varepsilon^{cr}$ is the critical strain. Equation (3) is expressed as buckling beam force–strain curves and Eqs. (4) and (5) are expressed as SMA wire tension–strain curves in Supplementary Fig. 8a. We compared the initial and final states of CASA derived from buckling beam force–strain curves and intersection points, where the buckling beam force–strain curves and SMA wire tension–strain curves (red and black) meet. The SMA wire model and buckling beam model can confirm the experimental observation on actuation of CASA with the different lengths of SMA wire. The values of these modeling parameters are reported in Supplementary Table 1.

Supplementary Fig. 1d represents an equivalent model for BPS. The bistability of the BPS is determined by the elastic beam stiffness $k_b$, parallelogram structure stiffness $k_p$, and height of the initial curved beam $h$. $k_b$ is calculated according to the PRB theory. $k_p$ is measured by the compression test. The total potential energy $U_{total}$ is expressed by the summation of the energies of the elastic beam $U_b$, and parallelogram structure $U_p$:

$$U_{total} = U_b + U_p \qquad (7)$$

$$U_{total} = 2 \cdot \frac{1}{2} k_b x^2 + \frac{1}{2} k_p (y - h)^2 \qquad (8)$$

Here, $x$, and $y$ are the angle of the torsion spring, deformation of the elastic beam, and vertical displacement of the BPS, respectively. From Eq. (8), we can estimate the energy–vertical displacement relationship, as shown in Fig. 3e. In addition, the reaction force of the BPS can be obtained as:

$$F_r = \frac{\partial U}{\partial y} = 2k_b x \frac{\partial x}{\partial y} + k_p(y - h) \qquad (9)$$

The reaction force–vertical displacement relationship of the BPS is shown in Fig. 3d.

**Performance characterization.** Supplementary Fig. 7a demonstrates the experimental setup for measuring the actuation stroke. To measure the actuation stroke of the CASA and position of the BPS, we placed the CASA and BPS under a laser displacement meter (Panasonic HG-C1030-P). Electrical voltage data measured by the laser displacement meter were acquired with DAQ (DEWESoft Korea, Ltd., SIRIUS). The CASA was placed in a tensile tester (Instron 3343) (Supplementary Fig. 7b) to measure its blocked force (Fig. 2e; Fig. 4g; and Supplementary Fig. 12d). Since The maximum actuation strain started to converge for an applied current above 0.26 A as shown in Supplementary Fig. 5a, we applied 0.28 A for 2 s to measure the actuation force applied on the upper block.

Similarly, the BPS was placed in the tensile tester to measure its continuous reaction force data, as shown in Fig. 3d. The BPS was bonded to the load cell tip to measure the reaction force due to the snap-through effect when the stage is both pulled and pushed.

To determine the power density of the CASA, we placed the actuator under a laser displacement sensor to measure its displacement and calculate velocity and acceleration. The CASA was actuated with loads ranging from 10 to 60 g to determine the maximum power density, as shown in Fig. 2f. Supplementary Fig. 7d plots the actuator response for a 40 g load for which the maximum power density was measured between the initial actuation, $t_i$, and the equilibrium state, $t_e$. The force applied by the CASA and specific power are calculated as:

$$F_{CASA} = m_{Load} \cdot (a + g) \qquad (10)$$

$$p = \frac{1}{m_{CASA}} F_{CASA} \cdot v \qquad (11)$$

where $F_{CASA}$, $m_{Load}$, $a$, and $g$ are the actuation force, mass of the applied load,

acceleration of the mass, and gravitational acceleration, respectively; further, $p$ and $m_{CASA}$ are the specific power and mass of CASA, respectively, and $v$ is the velocity. The peak of the specific power density can be calculated from Eq. (11), as shown in Supplementary Fig. 7e. To calculate the average specific power, the total specific work is determined and divided by the change in time between the initial actuation $t_i$ and equilibrium state $t_e$.

$$p_{avg} = \frac{1}{m_{CASA}(t_e - t_i)} W \qquad (12)$$

Supplementary Fig. 12b illustrates a few sample experimental measurements of the actuation stroke and input voltage of the CASA, with and without BPS. To quantify the input energy, we measured the input voltage ($V$) and corresponding current ($I$) at each actuation frequency from 0.1 to 1.25 Hz for 20 s as shown in Supplementary Fig. 12c. The total input energy in each frequency is:

$$E_{in} = \int_0^T V(t)I(t)dt \qquad (13)$$

Supplementary Fig. 12e demonstrates the measured electrical energy input. The input energy of the CASA without BPS is equal in all frequency ranges, because it consumes this energy during the actuating period for sustaining the actuation stroke. However, as shown in Fig. 3f, the actuation stroke of the CASA without BPS decreases sharply with increasing frequency, due to which the CASA without BPS does not achieve the required actuation stroke in high frequency.

To quantify the output work, we measured the actuation stroke $d$, average force $F_{avg}$, and number of actuations $n$ at each frequency for 20 s. To characterize the average force of the CASA without BPS, we measured the blocked force, as shown in Supplementary Fig. 11d. The average force of the CASA with BPS is the average of the measured effect until the snap-through effect is observed, as presented in Fig. 3d. The total mechanical output work (Fig. 3g) in each frequency is:

$$E_{out} = nF_{avg}d \qquad (14)$$

The energy efficiency in each frequency with and without BPS can be estimated by the following equation.

$$\eta = \frac{E_{out}}{E_{in}} = \frac{\text{Output work}}{\text{Input energy}} \qquad (15)$$

**Cyclic test for temperature evaluation of SMA in CASA.** The weight of the SMA wire embedded in the CASA is in sub-milligrams, which means the thermal capacity of the SMA wire is small by proportional to the weight. The thermal energy required by the SMA wire is not high enough to affect the environmental temperature in our cases. To prove this, we observed the temperature of the CASA under 1600 cycles of activation to determine if heat accumulation occurred in the SMA. To observe the temperature of the SMA wire in the CASA, we used an infrared camera (A655SC, FLIR, Co., Gangnam-Gu, South Korea), as shown in Supplementary Fig. 2 and Movie 3. During the cyclic actuation of the CASA, the heat of the actuating SMA wire does not accumulate to exceed a certain temperature limit (~70 °C). The sample length, actuation time, applied voltage and current of the SMA wire are presented in Supplementary Table 6.

**AR glasses prototype.** VAC occurs when our brain perceives mismatching between the distance of a virtual object (vergence) and the focusing distance for the eye to focus on that object (accommodation) (Supplementary Fig. 16a). The lens (or mirror) equation, $d_i = fd_0/(d_0 - f)$, implies that the image distance can be adjusted by varying the focal length of the lens or the distance of the object (where $d_i$, $d_0$, and $f$ are the distance between the lens and image, distance between the object and lens, and focal length of the lens, respectively). Compared to previously proposed varifocal display architecture[56,57], "binary focus" switching using the CASA has limited focus plane. The primary purpose of implementation of the CASA to AR glasses prototype is reducing form factor while maintaining high image quality and large field of view. To build millimeter-sized actuator with high power density, focus planes are intentionally limited. At the same time, considering arm's-length display interaction, we define the switching boundary between near and far depth as ~2 D[56]. CASA switches the distance between display and optics (mirror) to match the distance of a virtual object and the focusing distance of the eye (Supplementary Fig. 16b) within 0.3 s (from far to near depth) and 0.5 s (from near to far depth), respectively (Supplementary Fig. 16c). This switching speed can be controlled by SMA diameter and applied current. Although the human eye accommodation speed varies with age, it is generally in the range 200–300 ms (5 Hz)[58], which is sufficiently covered by CASA's actuation (~1.5 mm of displacement under a frequency of 5 Hz). The prototype implements two depths, near (3.3 D) and far (0.2 D). The required depth of an image can be determined by the contents. For 2D contents, suitable depths of contents are selected according to contents size. For 3D contents, rendering algorithm for binocular parallax is added. For instance, it is assumed that a 3D object is moving from a far depth to a near depth. In the case of general single depth AR glasses using only binocular parallax, a vergence is adjusted solely due to rendering algorithm regardless of the virtual image distance. On the other hand, in the case of the proposed AR glasses, when the object is at a far depth, CASA maintains the off state. When an 3D object

reaches a certain distance (2 D), CASA actuates and reduces the distance between the display and the optics. The binocular parallax is adjusted according to the distance by an algorithm (Supplementary Fig. 16d), and at the same time, the position of the image is also adjusted to resolve VAC. According to this interaction, VAC is able to be relieved since vergence and accommodation can be matched. The image switching due to the depth adjustment implemented by the CASA was captured by a CCD camera (FLIR Flea3) and a short depth-of-focus lens (12 mm/ f1.4) (Fig. 4e). The slight magnification changes in the 3D object, caused by a distortion induced by the relative distance between the display and combiner, is calculated beforehand and compensated using an image correction algorithm (Supplementary Fig. 16e).

**Tactile test design**. To evaluate the identification rate of the different contact forces, we conducted an experiment to measure signal intensity perceived by the user. All experiments on human were conducted under the IRB protocol (IRB No. 202203-HB-EX-001) approved by Ajou University Institutional Review Board. We recruited healthy adults as participants ($n = 12$, 9 male and 3 female; age $= 26.8 \pm 1.9$). We informed participants of risks and benefits prior to the experiment and obtained their consent. The subjects received an appropriate participant monetary compensation. Two types of tests were conducted in the laboratory under the same conditions. To minimize possible effect caused by a visual signal, participants were blindfolded. The user then randomly perceived one of four different magnitudes of the forces or the 'off' state in each test and were asked to identify the stimulus. Each contact force and off state were randomly applied for 10 times (50 stimuli in total for each test).

**Statistics and reproducibility**. No statistical method was used to predetermine sample size. No data were excluded from the analyses. The experiments were not randomized. The Investigators were not blinded to allocation during experiments and outcome assessment. Microsoft Excel was used for data analysis.

**Reporting summary**. Further information on research design is available in the Nature Research Reporting Summary linked to this article.

## Data availability
Source data for the main figures are provided with this paper. Other raw and analyzed datasets generated during the study are available for research purposes from the corresponding authors on reasonable request. Source data are provided with this paper.

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

## Acknowledgements

This work is funded by the Samsung Advanced Institute of Technology and supported by Samsung Advanced Institute of Technology, the new faculty research fund of Ajou University, the Ajou University research fund, and Basic Science Research Program through the National Research Foundation of Korea (NRF-2021R1C1C1011872, 2019R1C1C1007629). The design of the AR prototype was supported by Corporate Design Center of Samsung Electronics. We acknowledge the advice and discussion about the demonstration of the haptic glove from D. Lee and all members of the Ajou Multiscale Bio-inspired Technology Laboratory for their help and assistance.

## Author contributions

D.Kim, B.K., B.S., D.Kang, S.Han, H.-S.L., and J.-S.K. initiated the project. D.Kim, B.K., B.S., and J.-S.K. designed and conducted the research. D.Kim, B.K., and D.S. fabricated the haptic glove. B.S., C.-K.L., G.S., W.S., and S.K. designed the optics. J.-S.C. and J.S. built the operation board. Y.-T.K. built a 3D rendering algorithm. D.Kim, B.K., B.S., S.Hong, H.-S.L., S.Hwang, and J.-S.K. provided preparation of the manuscript and feedback. S.Han, D.Kang, H.-S.L., and J.-S.K. supervised the work.

## Competing interests

The authors declare no competing interests.
