## [Peer Review File · Nature Communications]

Actuating compact wearable augmented reality devices by multifunctional artificial muscleREVIEWER COMMENTS

Reviewer #1 (Remarks to the Author):

The authors present a shape memory alloy driven actuator (CASA) with bistable mechanism (BPS) for application in augmented reality devices. Comprehensive characterization of the SMA actuator is performed. Although the application of this SMA actuator in an AR device is novel, the design of this SMA actuator (strain-amplification mechanism) has been presented by the author in [36]. Moreover, driving multiple loops of thin SMA wires to achieve equivalent force output at higher actuation frequency by high voltage and low actuation time has been shown in many works (e.g. "SMALLBug: A 30-mg Crawling Robot Driven by a High-Frequency Flexible SMA Microactuator" by Nguyen et al. and "SMARTI: A 60-mg Steerable Robot Driven by High-Frequency Shape-Memory Alloy Actuation" by Bena et.al). The following concerns also need to be addressed.

1. For Fig. 1c, the author only includes some of the commercial SMA actuators but neglect many research works that have shown sma actuators with high power to weight ratio (e.g. "SMALLBug: A 30-mg Crawling Robot Driven by a High-Frequency Flexible SMA Microactuator" by Nguyen et al.)
2. It is not scientific to highlight a high strain (300%) with external preload (80g). The strain should be reported without external preload (56.1%). The high strain is also reported in the Abstract and Introduction without presenting the condition (external preload), which is misleading to readers.
3. For the characterization results presented in Fig. 2d-i, the sample length, actuation time, actuation voltage and current should be reported for SMAs with a diameter of 0.025mm, 0.05mm and 0.1mm respectively in the main manuscript. Although using multiple loops of thin SMAs can result in higher bandwidth with equivalent output force, more energy may be required for each actuation due to larger surface areas when compared with fewer loops of thick SMAs. The intervals and range of X and Y axis in Fig. 2g and 2h should be presented in detail.
4. It is obvious that the bistable mechanism can enhance the energy efficiency of the CASA at low actuation frequency. How about high frequency? What is the minimum power requirement to drive the CASA with the BPS? How to precisely control the actuation time for the CASA with BPS? ensure the actuation without wasting extra energy compared with the CASA without the BPS? What is the force required from the SMAs to trigger the bistable actuation from the SMA embedded in CASA and the external SMAs respectively? Is the power consumption from the external SMAs to trigger the actuation toward the other side included in the calculation and Fig. 3h? It seems not according to the voltage plot shown in supplementary Fig. 11b. For Fig. 3f, 20s of actuation at each frequency is not enough to guarantee that the actuation amplitude converges to a stable value. At least 50-100 cycles should be actuated at each frequency. You can report the results with fewer actuation frequencies.
5. Sample length, actuation time, actuation frequency, actuation voltage and current of the SMAs embedded in CASA and externally used in the AR devices should be reported respectively in detail in the main manuscript. Instead of presenting in the Methods section, the detail of the switch from 3.3D to 0.2D with the actuation of the counter SMA wires should also be reported in the "Actuating Compact AR devices" subsection.
6. Discussion section is too short. At least future work and limitation of this works should be discussed.
7. In the Methods section, the authors mention about the actuation time for blocking force test is 2s. However, the actuation time is much shorter in the application of the actuator. What is the blocking force at the actuation frequency the author presents?
8. In the "AR glasses prototype" subsection, the authors claim that "Although the actual human eye accommodation speed varies with age, it is generally 200~300 ms (5 Hz)⁴², which is sufficiently covered by CASA's actuation (~30 Hz)". However, the actuation frequency in the AR glasses prototype presented in this paper is much lower. The actuation amplitude at 30 Hz is too small to be useful in any AR device. So this is not an appropriate claim.
9. Sample length, actuation time, actuation frequency, actuation voltage and current of the SMAs embedded in CASA and externally used in the AR devices should be reported respectively in detail for the cyclic test in the "Cyclic test for temperature evaluation of SMA in CAS" subsection.
10. A DMA test is highly recommended for the SMA actuators to find out the corresponding transition temperature (A_s , A_f , M_s and M_f), force and strain and optimize the operation condition correspondingly.

11. A significant limitation of this work is that the actuation is bistable (on and off). Precise control of the strain inside the range is not presented, which limits the application of this actuator in AR application.

Reviewer #2 (Remarks to the Author):

This manuscript addresses a really interesting topic of designing light-weight and high-power actuator based on Shape Memory Alloy (SMA) combined with strain-amplification.

Overall, manuscript is well written and supported with supplementary materials including a number of video clips that demonstrate various aspects of the actuator.

I am only able to provide review from my expertise related to on-body haptic feedback using SMA actuators.

Recent work demonstrates that SMA actuators are particularly suited to physically deform the skin because of their large displacement length and high force-to-weight ratio (<https://dl.acm.org/doi/10.1145/3290605.3300718>). Would be good to cite this work.

Also, it would be good to understand the amount of expressivity (types of feedback) that can be generated by the proposed actuator or combination of multiple actuators. There was a very related work that looked into recreating various touch gestures using a matrix of wearable SMA patches (<https://dl.acm.org/doi/abs/10.1145/3313831.3376491>).

In addition, the adaption of SMA actuators for haptic applications such as immersive VR is still at an early stage, especially because it requires VR developers to understand the complex non-linear behaviour of SMA-based actuators. Recent work, e.g. <https://dl.acm.org/doi/10.1145/3411764.3445613>, have proposed new approach that allows for easy fabrication of SMA based actuators on cloths. This could be a useful point to include in the discussion/future directions.

Given my expertise in designing body-based interactions, I believe this will inspire and stimulate a lot of discussion in on-body/wearable space. Also, I could see the value of the proposed actuator influencing many applications that require on-body haptic feedback.

Reviewer #3 (Remarks to the Author):

This paper describes a type of SMA wire-driven actuator design for applications in augmented reality (AR). More specifically, a Compliant Amplified SMA actuator (CASA) is designed to amplify the small strain of SMA wire and a Bistable Parallelogram Linear State (BPM) is designed to further improve its energy efficiency. Two demonstrations are performed, one as a focus adjusting mechanism in a VR glass, and the other as a static force simulating mechanism worn on human's fingers. Overall, this paper is interesting and written in great detail. Some key parameters regarding actuators and devices are carefully investigated. The performance of the actuators is great and the supplementary videos show a very promising technology. However, the current version does not contain sufficient description of their novelty in terms of actuator design and applications. The following questions need to be addressed before considering publication in Nature Communications.

1. The authors need to clearly define some of their key words earlier in their paper. For example, the "small form factor". How thick should the device be and how do we define a small form factor? Also, how fast should the focus adjusting system be? If it is to be done within 300 ms, it seems like the frequency of the actuator has no need to reach 30 Hz.

2. CASA actuator. The CASA mechanism proposed in this paper is similar to other Strain Amplification Mechanisms, both for SMA wire actuators and other types of actuators [1] [2]. The

authors need to claim their uniqueness of their design to state CASA is one of their important contributions.

3. CASA actuator power density. The calculated power density of CASA is up to 1.7 kW/kg. This is definitely an amazing result. However, after further checking the displacement, force, as well as dynamic properties of CASA, the reviewer suspects this data might be incorrectly calculated. The authors need to recalculate it. It is true that the displacement and force, and thus energy density of CASA is great, yet its properties drop down quickly as frequencies increases. Thus, at high frequency like 30 Hz, both the force and the strain are small, ending up with a limited power density. If it is 1.7 kW/kg, the authors need to clearly describe at which frequency have they calculate the 1.7 kW/kg power density, as well the generated force and displacement related to this frequency on page 7. Also, the force vs frequency data need to be supplemented in Fig. 2d.

4. How CASA and BPS are combined should be described.

5. The demonstrations of both the focus adjusting and the static force simulating lack quantitative evaluation to support the superiority of their design. The authors claim their actuators are of high speed, then the time response of their final integrated device should be clearly shown. Also, how does the user feel about wearing the AR glass with such a focus adjusting mechanism? Does it really release fatigue for the users? These need to be quantitatively evaluated. Human tests need to be done to verify the functionality of the haptic glove design as well, as sometimes, giving some amount of force on the hand, does not necessarily mean the wearers feel like they are touching something. The reviewers believe more in-depth evaluation of the two haptic devices need to be conducted, to prove the superiority of their proposed device.

[1] Ameduri, Salvatore, Angela Brindisi, Monica Ciminello, Vincenzo Quaranta, and Marco Brandizzi. "Car Soundproof improvement through an SMA Adaptive system." In *Actuators*, vol. 7, no. 4, p. 88. Multidisciplinary Digital Publishing Institute, 2018.

[2] Ueda, Jun, Thomas W. Secord, and H. Harry Asada. "Large effective-strain piezoelectric actuators using nested cellular architecture with exponential strain amplification mechanisms." *IEEE/ASME transactions on mechatronics* 15, no. 5 (2009): 770-782.

Response to Reviewer #1 Comments

Comment 1: The authors present a shape memory alloy driven actuator (CASA) with bistable mechanism (BPS) for application in augmented reality devices. Comprehensive characterization of the SMA actuator is performed. Although the application of this SMA actuator in an AR device is novel, the design of this SMA actuator (strain-amplification mechanism) has been presented by the author in [36]. Moreover, driving multiple loops of thin SMA wires to achieve equivalent force output at higher actuation frequency by high voltage and low actuation time has been shown in many works (e.g. “SMALLBug: A 30-mg Crawling Robot Driven by a High-Frequency Flexible SMA Microactuator” by Nguyen et al. and “SMARTI: A 60-mg Steerable Robot Driven by High-Frequency Shape-Memory Alloy Actuation” by Bena et.al). The following concerns also need to be addressed.

Our response: We thank the reviewer for their comment. The main query is regarding is the novelty of our work compared to other studies. The strain-amplification technique is a well-known method that is widely used to develop piezoelectric and SMA actuators. Because our design is based on the fundamental mechanism of the strain-amplification technique, we understand that a similar schematic of the SMA actuator presented by the author in [40] could raise concerns about the novelty of our work.

However, we believe the advances of our work in terms of the modelling and material-based design with respect to the state of the art have enabled novel AR and wearable applications with a high power, compact form factor, and high energy efficiency that other strain-amplified actuators cannot fully achieve.

We respect the previous work [40] by introducing the design of the bow-like SMA actuator and captures the relationship between the force and stroke of the actuator with analytical modelling. However, the previous design is not practical because its power density is expected to be very low owing to its large weight. As described in our manuscript, SMA actuators inevitably require additional components that weigh much more than the bare SMA, causing a decrease in the power and energy density by an order of magnitude, limiting their applications. Minimising the additional parts while achieving the desirable function of the SMA device is necessary. Our actuator was designed to minimise additional components while joining two compliant beams and an SMA wire. The CASA consists only of the SMA wire, small crimps,

and two compliant beams that are precisely patterned to join each other, working as a hinge by themselves. In contrast, the prototype in the previous work consisted of not only compliant beams and SMA wires but also bulky blocks, nuts, ferrules, headless screws, and a telescopic guide, resulting in a large volume and weight. These restrict the power density of the actuator as well as scalability, limiting its applications, especially in AR and wearable devices, where high power density, small volume, and weights are required for the constraints.

Moreover, we established an optimised model for combining an SMA wire and compliant beams to determine the optimal length of the SMA wire. Determining the optimal length of the SMA wire is critical for the practical usage of strain-amplification mechanism-based devices because it determines the actuation strain as well as the initial volume (height). We achieved the highest actuation strain under the given conditions with the derived pseudo-rigid-body model of the compliant beam and a one-dimensional nonlinear SMA model. The analytical models are simple to use but precise and accurate for capitalizing on the artificial muscle with strain-amplification mechanism to the maximum

SMA actuators have an inherently low energy efficiency. Although power consumption is one of the most important issues for practical usage, it has not been sufficiently addressed in many SMA-based devices. We developed a bistable parallelogram linear stage (BPS) to compensate for low energy efficiency and tilting. These critical problems of SMA-based actuators are addressed by integrating the CASA with a single sheet of BPS which allows for a low-profile and compact form factor of the actuator while achieving a high energy efficiency.

In addition, the CASA has great potential as a multifunctional actuator capable of actuation and sensing isomorphically. We exploited the resistance-changing characteristic of the SMA in the CASA to utilise it as a force sensor, which was not included in our previous manuscript. As shown in Fig. 6, we employed CASA as a resistance-type force sensor to measure the external contact and added a sensing function to the haptic glove for tactile communication without embedding separate actuators and sensors in the glove. The details of the sensing capability are described in the revised manuscript and Supplementary Movie 8. We believe that the additional function of the CASA as a sensor could support the novelty of our work.

Revised manuscript:

(page 4 paragraph 10) Previous studies^{21, 40} exploited the mechanism used to amplify the small strain of the SMA and the piezoelectric actuator. In contrast, we established an optimizing design of the CASA to achieve relatively high actuation strain (56.1%), power density (1.7 kW/kg), and thin form factor (5 mm of height) with the desirable force and frequency for practical use in AR wearable applications. The CASA also combines an SMA wire and a compliant structure that amplifies the strain without heavy rigid components required for amplification mechanism in the previous study.

(page 5 line 13) Last, we utilized the CASA as an resistance-type force sensor to detect external contact. The CASAs provide both sensing and actuation function without embedding separate actuators and sensors in the haptic devices.

(Page 13-15) Sensing capability for tactile communication

In addition to actuation, the sensing function is highly desirable in wearable systems^{43, 44}. The tight integration of the actuation and sensing⁴⁵⁻⁴⁷ provides novel multifunctional devices increasing application potential for next-generation wearable systems. The CASA with the embedded smart material embedded has a favorable characteristic to couple sensing and actuation for applications such as communication for deaf-blind people through the haptic glove.

The electrical resistance of the SMA can be varied by changing the strain. The SMA-based device can be controlled through resistance-based self-sensing techniques^{48, 49}. Figure 6a shows that the CASA can be utilized as a resistance-type force sensor for measuring the external contact. Before pressure is applied, the compliant bending beams of CASA and routed SMA wire with constant resistance ($R1$) are in equilibrium. External pressure on the CASA causes the bending compliant beams to straighten, resulting in the extension of the SMA wire. The resistance of the SMA wire in CASA, thus, increases ($R2$). Once the external pressure is removed, the compliant beams and the SMA wire are restored to their original states, and the resistance also returns to its original value ($R1$).

Figure 6b shows the loading and unloading characteristics of the two CASAs with different diameters of SMA wire (0.1 mm and 0.025 mm). The CASA with the 0.025 mm diameter SMA wire and 0.1 mm-thick compliant beams exhibits a relatively high sensitivity of 0.1498 N^{-1} under an applied force of 0.13 N. The CASA with the 0.1 mm diameter SMA wire and 0.2

mm-thick compliant beam exhibits a sensitivity of 0.01595 N^{-1} under an applied force of 1.1 N. The sensitivity and measurable force range are determined by the beam thickness and wire diameter. The high stiffness of the CASA with the thick beam (0.2 mm) and thick wire (0.1 mm) enables a wide range of measurable force, while the low stiffness of the CASA with the thin beam (0.1 mm) and thin wire (0.025 mm of diameter) provides high sensitivity. During unloading, the response characteristic aligns with the corresponding loading characteristic without noticeable hysteresis under the applied force. The results of the loading–unloading test conducted for approximately 1000 cycles demonstrate the durability of the CASA as a force sensor (Supplementary Fig. 20).

The CASA exhibiting sensing capability and high actuation performance can be used to develop highly compact haptic gloves for the tactile communication described in Fig. 6c without embedding separate actuators and sensors (Supplementary Movie 8). Tactile communication allows deaf-blind people to communicate through a Braille-based device⁵⁰⁻⁵². A Braille cell consisting of six dots that can represent letters, numbers, and special characters with the combination of raised dots⁵². Through the finger Braille communication method, the user can represent the six dots of Braille using the index, middle, and ring fingers of both hands⁵¹.

As shown in the scenario in Fig. 6c, users are wearing the haptic gloves with the CASAs embedded on the index (L3, R3), middle (L2, R2), and ring (L1, R1) fingers of both hands. User A presses the corresponding fingers on the ground to represent the Braille code as shown in Supplementary Fig. 21. The CASAs embedded on each finger are also applied by contact force and the letter represented by the Braille code can be identified based on the resistance changes in the SMA wire in the CASAs. The resistance data of the CASAs can be used to actuate the corresponding CASAs embedded in user B's haptic glove, allowing user B to receive the message. User B can send messages to user A in the same way.

Figures 6d and 6e shows the letters H, O, W, A, R, E, Y, O, U, ? and G, R, E, A, T, ! are typed by users by tapping the appropriate combination of the fingers on the desk with the haptic glove. The resistance changes in the CASAs as sensors are clearly identified for each letter. The previously demonstrated actuation characteristic of the CASA enables the user to sense received messages as well.

Fig. 6 | CASA with sensing capability and its application to tactile communication. a, Schematic of CASA’s sensing capability measuring the external contact. **b,** Normalized resistance as a function of applied force under loading–unloading test. **c,** Concept of tactile communication using actuation and sensing capability of the CASA. **d** and **e,** Normalized sensor output for each finger and corresponding Braille code to input H, O, W, A, R, E, Y, O, U, ? and G, R, E, A, T, !, respectively.

Supplementary Figure 20. Cyclic test results of loading–unloading loop for 5,000 s. a, 0.1 mm diameter of the SMA wire. **b,** 0.025 mm diameter of the SMA wire. **c,** Cyclic test results of 0.1 mm diameter of the SMA wire for 10 cycles **d,** Cyclic test results of 0.025 mm diameter of the SMA wire for 10 cycles. Cyclic test speed on **(a)** and **(b)** are 50 and 40 mm/min, respectively.

Supplementary Figure 21. CASAs embedded in the haptic glove for actuation and sensing capability. CASAs located in L1, L2, L3, R1, R2, and R3 are corresponded to each Braille code.

Comment 2: For Fig. 1c, the author only includes some of the commercial SMA actuators but neglect many research works that have shown sma actuators with high power to weight ratio (e.g. “SMALLBug: A 30-mg Crawling Robot Driven by a High-Frequency Flexible SMA Microactuator” by Nguyen et al.)

Our response: We thank the reviewer for providing us with an additional reference to show the wide range of power densities and weights of the SMA-based actuators. We have added a reference [22] to provide and calculate the power density of the actuator for comparison, as shown in Fig. 1c. According to reference [22], the mass of the actuator is 6 mg, and the load is 83.83 mN 1 Hz. The velocity is 17.24 mm/s (1 mm of the displacement and actuation time of 78 ms), respectively. Based on these values, the power density was calculated to be 240 W/kg which is higher than that of the other SMA-based actuator previously included in Fig. 1c. This is a good reference to demonstrate the light weight and high power density of SMA-based actuators.

Revised manuscript:

Fig. 1 | Overview of wearable devices using the CASA in an AR environment.

Revised manuscript:

(Page 3 line 13) *Smart materials such as piezoelectric transducers (PZTs)¹⁹⁻²¹, shape memory alloys (SMAs)²²⁻²⁴, and dielectric elastomer actuators (DEAs)²⁵⁻²⁸ are likely candidates (Fig. 1c).*

Comment 3: It is not scientific to highlight a high strain (300%) with external preload (80g). The strain should be reported without external preload (56.1%). The high strain is also reported in the Abstract and Introduction without presenting the condition (external preload), which is misleading to readers.

Our response: We agree that the report can be misleading to the readers. The power of the actuators changed depending on the payload. We have thus modified our manuscript accordingly.

Revised manuscript:

(Page 2 line 4) *Despite its light weight (0.22 g), the CASA has a high power density of 1.7 kW/kg and an actuation strain of 300% under 80 g of external payload.*

(Page 4 line 18) *The actuator weighs 0.22 g but has a maximum actuation strain of 300% under 80 g of external payload, a payload 800 times higher than its weight, and a high power density of 1.7 kW/kg (> 0.2 kW/kg for biological muscle).*

Comment 4: For the characterization results presented in Fig. 2d-i, the sample length, actuation time, actuation voltage and current should be reported for SMAs with a diameter of 0.025mm, 0.05mm and 0.1mm respectively in the main manuscript. Although using multiple loops of thin SMAs can result in higher bandwidth with equivalent output force, more energy may be required for each actuation due to larger surface areas when compared with fewer loops of thick SMAs. The intervals and range of X and Y axis in Fig. 2g and 2h should be presented in detail.

Our response: We thank the reviewer for this comment. The sample length, actuation time, actuation voltage, and current are presented in Supplementary Table 6 and referred to it in the manuscript. Fig. 2g and 2h show conceptual graphs that more clearly express the characteristics of the SMA and compliant beams. The actual results are presented in Supplementary Fig. 8a and 8b.

	Sample length	Actuation time, Cooling time (Frequency)	Actuation voltage	Actuation current
Figure 2d	50 mm (Diameter of 0.1 mm)	50 ms, 9950 ms (0.1 Hz)	9.5 V	1.5 A
		50 ms, 1950 ms (0.5 Hz)	9.5 V	1.5 A
		50 ms, 950 ms (1Hz)	8 V	1.2 A
		50 ms, 617 ms (1.5 Hz)	7 V	1.1 A
		50 ms, 450 ms (2 Hz)	5.9 V	0.93 A
		50 ms, 350 ms (2.5 Hz)	5.4 V	0.85 A
		50 ms, 283 ms (3 Hz)	5 V	0.8 A
	50 mm (Diameter of 0.05 mm)	10 ms, 1990 ms (0.5 Hz)	9.5 V	0.38 A
		10 ms, 990 ms (1 Hz)	9.5 V	0.38 A
		10 ms, 490 ms (2 Hz)	8.7 V	0.352 A
		10 ms, 323 ms (3 Hz)	7.8 V	0.31 A
		10 ms, 240 ms (4 Hz)	7 V	0.28 A
		10 ms, 190 ms (5 Hz)	6.5 V	0.26 A
		10 ms, 157 ms (6 Hz)	6 V	0.24 A
		10 ms, 133 ms (7 Hz)	5.7 V	0.23 A
		10 ms, 115 ms (8 Hz)	5.4 V	0.22 A
	101 mm (Diameter of 0.025 mm)	10 ms, 995 ms (1 Hz)	25 V	0.18 A
		10 ms, 195 ms (5 Hz)	21 V	0.15 A
		10 ms, 95 ms (10 Hz)	18 V	0.13 A
		10 ms, 62 ms (15 Hz)	15.5 V	0.11 A
		10 ms, 47 ms (20 Hz)	13.5 V	0.1 A
10 ms, 38 ms (25 Hz)		12.5 V	0.09 A	
10 ms, 32 ms (30 Hz)		12.2 V	0.09 A	
Figure 2e	50 mm (Diameter of 0.1 mm)	2 sec (actuation time)	1.78 V	0.28 A
	101 mm (Diameter of 0.1 mm)		3.6 V	
Figure 2f	50 mm (Diameter of 0.1 mm)	20 ms (actuation time)	13 V	2.1 A
	101 mm (Diameter of 0.1 mm)		20 V	3.15 A
Figure 2i	48 mm	2 sec (actuating time)	1.75 ~ 1.8 V	0.28 A
	49 mm			
	50 mm			
	51 mm			
	52 mm			
AR device (CASA with BPS)	18 mm (SMA in CASA)	40 ms (actuating time)	3 V	0.5 A
	27 mm (Counter SMA)	45 ms (actuating time)	4 V	0.5 A

Supplementary Figure 2	50 mm (Diameter of 0.1 mm)	1 sec (actuating time)	1.78 V	0.28 A
----------------------------	------------------------	--------	--------

Supplementary Table 6. Sample length, actuation time, voltage and current of SMA wire.

Revised manuscript:

(Page 8 line 16) *The sample length, actuation time, applied voltage and current of the SMA wire in each test are presented in Supplementary Table 6.*

Comment 5

Our response: We thank the reviewer for detailed comments regarding BPS. We regret providing insufficient information on the BPS and its actuation characteristics, particularly with respect to the input electric energy and efficiency. We have added explanations of each reviewer's comment in the response and supplementary information with an additional experiment.

- **Comment 5-1:** It is obvious that the bistable mechanism can enhance the energy efficiency of the CASA at low actuation frequency. How about high frequency?

Our response: Even at a high actuation frequency, the CASA with BPS does not show a considerable decrease in efficiency. The actuation time of the CASA with the BPS is reduced by applying an impulsive electric current. This impulsive actuation enables a fast state transition between stable states. To verify the possibility of impulsive actuation, we performed a parametric study on the input electric current and actuating time, as shown in Supplementary Fig. 13. In the experiment, the assigned actuating time is the minimum value corresponding to the applied electric current. The experimental results show that increasing the electric current induces a fast transient time under 200 ms, which is sufficiently fast for adjusting the accommodation of the human eye, and the value is constant at various actuation frequencies. Therefore, the input energy for a single actuation becomes constant, and the efficiency can be maintained at various frequencies.

Supplementary Figure 13. parameter study on current and actuating time. a–g. Applied voltage and actuation stroke with changing electric current and actuating time (actuating time of CASA and counter SMA wire are denoted in blue and yellow, respectively.).

- **Comment 5-2:** What is the minimum power requirement to drive the CASA with the BPS?
- **Comment 5-3:** How to precisely control the actuation time for the CASA with BPS to ensure the actuation without wasting extra energy compared with the CASA without the BPS?

Our response (Comment 5-2 and 5-3): The electric power applied to the CASA and counter SMA wire can be calculated from the average voltage and current, as shown in Supplementary Fig. 14a. The minimum power required for driving the CASA and counter SMA wire are 0.194 W (0.779 V and 0.25 A) and 0.297 W (1.19 V and 0.25 A), respectively. However, the total energy input for actuation gradually decreases by applying an impulsive electric current in the CASA and counter SMA wires, as shown in Supplementary Fig. 14b and c. Through this parameter study, we can respond to the reviewer’s comment regarding the precise control of the actuation time. As shown in

Supplementary Fig. 13, the state transition time of the BPS is proportional to the applied voltage time. Impulsive actuation has advantages in terms of not only the rapid transition of the state but also the minimum electric energy. Supplementary Figure 14b and 14c show the input energies calculated from the applied voltage and current, respectively. When a high current is applied for a short time, the input energy is close to the theoretical heat energy. the amount of theoretical heat required to increase the temperature was calculated as follows:

$$Q = mc_p\Delta T$$

where Q is the quantity of energy, m is the mass of the SMA wire, c_p is the specific heat capacity of the SMA wire, and ΔT is the increase in temperature of the SMA wire. The dimensions and thermal properties of the SMA wires are listed in Supplementary Table 5. By comparing the calculated input energy and theoretical heat energy, we confirmed that adjusting the actuation time can reduce the wasted energy caused by heat convection. Moreover, the BPS with the CASA maintained its position without electric energy. This means that the energy consumption to maintain near or far depth in AR applications is zero.

Supplementary Figure 14. Input power and energy with varying electric current and actuation time. a, CASA input power with changing current and actuating time. b, CASA input energy with changing current and actuating time. c, Counter SMA wire input energy with changing current and actuating time.

Parameter	Symbol	Value
Weight of SMA wire in CASA	m_{CASA}	0.7852 mg
Weight of counter SMA wire	$m_{counter}$	1.2563 mg
Specific heat of SMA wire	c_p	0.836 J/g · °C
Density of SMA wire	ρ_{SMA}	6.45 g/cm ³
Diameter of SMA wire	W_{hinge}	0.1 mm
Length of SMA wire in CASA	l_{CASA}	15.5 mm
Length of counter SMA wire	$l_{counter}$	24.8 mm
Room temperature	T_{room}	25 °C
Target temperature	T_{target}	90 °C
Theoretical heat energy of SMA wire in CASA	Q_{CASA}	42.67 mJ
Theoretical heat energy of counter SMA wire	$Q_{counter}$	68.27 mJ

Supplementary Table 5. Thermal characteristics of SMA wire.

(Supplementary Materials page 2)

Supplementary Text 2. Impulsive actuation and consumed energy of BPS

The actuation time of CASA with BPS is reduced by applying an impulsive electric current. This impulsive actuation enables a fast state transition between stable states. To verify the possibility of impulsive actuation, we performed a parametric study on input electric current and actuating time, as shown in Supplementary Fig. 11. The electric power applied to the CASA and counter SMA wire can be calculated from average voltage and current, as shown in Supplementary Fig. 12a. Although the power increases upon increasing the electric current, the total energy input for actuation gradually decreases upon applying impulsive electric currents in the CASA and counter SMA wire, as seen from Supplementary Fig. 12b and c.

The state transition time of BPS is proportional to the applied voltage time. The impulsive actuation has the advantages of rapid transition of state and minimum electric current. Supplementary Figure 12b and c show the input energy calculated from the applied voltage

and current. As the high current is applied for a short time, the input energy is close to the theoretical heat energy. Here, the amount of theoretical heat required to raise the temperature is calculated follows:

$$Q = mc_p\Delta T$$

where Q is the amount of energy, m is the mass of the SMA wire, c_p is the specific heat capacity of the SMA wire, and ΔT is the temperature rise of the SMA wire. The dimension and thermal properties of the SMA wire are described in Supplementary Table 5. Comparing the calculated input energy and theoretical heat energy, we confirm that adjusting the actuation time can reduce energy wastage due to heat convection. Moreover, the CASA with BPS maintain its position without electric energy. This means that no energy is consumed to maintain near or far depth in AR applications.

- **Comment 5-4:** What is the force required from the SMAs to trigger the bistable actuation from the SMA embedded in CASA and the external SMAs respectively?

Our response: In the bistable mechanism, the force required to switch its state is the local maximum and local minimum value of the force–displacement relationship. Thus, the force required by the CASA and the counter SMA wire to trigger the bistable actuation of the BPS are theoretically equal (approximately 0.5 N), as shown in Fig. 3d.

- **Comment 5-5:** Is the power consumption from the external SMAs to trigger the actuation toward the other side included in the calculation and Fig. 3h? It seems not according to the voltage plot shown in supplementary Fig. 11b.

Our response: The power consumption and energy efficiency in Fig. 3h include the energy consumed by both the CASA and counter SMA wire. We regret providing confusing information in the plot. Different colours are used in the plot to distinguish the two antagonistic actuators. In addition, the voltage plot in Supplementary Fig. 11b shows the actuation displacement (h) and applied voltage (V) of 0.1 Hz cyclic actuation to show the zero holding current of the actuator module, including the BPS. We have added the

entire voltage data corresponding to Fig. 3f, g, and h in the Supplementary Information, as shown in Supplementary Fig. 12c.

Supplementary Figure 12. Performance characterization of BPS. **a**, Experimental setup for measuring the reaction force of BPS. **b**, Sample experimental measurement (0.1 Hz) in Fig. 3f of actuation stroke and input voltage of the CASA with and without BPS. **c**, Actuation voltage

of the CASA with BPS and Counter SMA wire (upper plot) and CASA without BPS (lower plot) with increasing the actuation frequency corresponding to Fig. 3f. **d**, Blocked force plot of CASA with one line of embedded SMA wire. **e**, Input energy (E_{in}) plots of CASA with (red) and without (black) BPS with actuation frequency varying from 0.1 to 1.25 Hz for 20 s.

(Supplementary Materials page 15) c, *Actuation voltage of the CASA with BPS and counter SMA wire (upper plot) and CASA without BPS (lower plot) with increasing actuation frequency corresponding to Fig. 3f.*

- **Comment 5-6:** For Fig. 3f, 20s of actuation at each frequency is not enough to guarantee that the actuation amplitude converges to a stable value. At least 50-100 cycles should be actuated at each frequency. You can report the results with fewer actuation frequencies.

Our response: We agree that 20 s of actuation at each frequency is not sufficient to guarantee the convergence of the actuation amplitude. To verify the convergence of the actuation amplitude, an additional experiment was conducted. As shown in Supplementary Fig. 15, the CASA and counter SMA wire were actuated more than 100 times with various actuation frequency from 0.1 1 Hz. Actuation displacement is maintained even during repetitive actuation at each frequency.

Supplementary Figure 15. Displacement vs. time for different actuation frequencies. Cyclic actuation of BPS with CASA and counter SMA wire by varying frequency, **a.** 0.1 Hz, **b.** 0.2 Hz, **c.** 0.25 Hz, **d.** 0.4 Hz, **e.** 0.5 Hz, **f.** 0.8 Hz, and **g.** 1 Hz.

(Supplementary Materials page 17) *Supplementary Figure. 13. Parametric study on current and actuating time. a–i. Applied voltage and actuation stroke with changing electric current and actuating time (actuating time of CASA and counter SMA wire are denoted in blue and yellow, respectively).*

(Supplementary Materials page 18) *Supplementary Figure. 14. Input power and energy with varying electric current and actuation time. a. CASA input energy with changing current and actuating time. b. CASA input energy with changing current and actuating time. c. Counter SMA wire input energy with changing current and actuating time.*

(Supplementary Materials page 19) *Supplementary Figure 15. Displacement vs. time with actuation frequency.* Cyclic actuation of BPS with CASA and counter SMA wire with varying frequency, **a.** 0.1 Hz, **b.** 0.2 Hz, **c.** 0.25 Hz, **d.** 0.4 Hz, **e.** 0.5 Hz, **f.** 0.8 Hz, and **g.** 1 Hz.

(Supplementary Materials page 31) *Supplementary Table 5. Thermal characteristics of SMA wire in the actuation module.*

Comment 6: Sample length, actuation time, actuation frequency, actuation voltage and current of the SMAs embedded in CASA and externally used in the AR devices should be reported respectively in detail in the main manuscript. Instead of presenting in the Methods section, the detail of the switch from 3.3D to 0.2D with the actuation of the counter SMA wires should also be reported in the “Actuating Compact AR devices” subsection.

Our response: We thank the reviewer for the constructive comment. We have added the information in Supplementary Table 6 (Comment 4). We also described actuation of the CASA and counter SMA wire in the “Actuating Compact AR devices” subsection.

Revised manuscript:

(page 11 line 22) *The CASA can restore to its original state by the compliant beams. However, the BPS requires an additional actuator to restore to its original state (from 0.2 D to 3.3 D) after the actuation of the CASA. We employed the counter SMA wire passing through holes located in the centre of the BPS and anchored at the external frame as shown in Supplementary Fig. 11a. To return BPS, the counter SMA wire contracts by applying an electric current and pulls the moving platform of the BPS. When the actuation force of the counter SMA wire exceeds the BPS energy barrier, the BPS returns (Supplementary Fig.11f).*

Comment 7: Discussion section is too short. At least future work and limitation of this works should be discussed.

Our response: We thank the reviewer for their comment. We have added the future work possibilities and limitations of our work in the Discussion section as follows:

Revised manuscript:

(page 16 line 7) *However, the low precision of the strain control and high energy consumption rate are inherent limitations of SMA-based actuator. By combining the resistive sensing feedback, the strain control of the SMA can be improved, which is advantageous for developing the multifunctional actuator capable of force sensing without additional strain sensors, as the biological muscle does. In terms of energy consumption of SMA-based actuators, the absolute value of energy consumption in micro-scale applications is low enough for battery use. Using a mechanical design with passive appendages, such as the bi-stable structures used in this study, also increases the working time of SMA-based actuators. In addition to wearable device applications, future work will attempt to develop a platform for on-textile interfaces⁵³ for clothing actuation keeping wearer comfort in mind.*

Comment 8: In the Methods section, the authors mention about the actuation time for blocking force test is 2s. However, the actuation time is much shorter in the application of the actuator. What is the blocking force at the actuation frequency the author presents?

Our response: We thank the reviewer for the thoughtful comments. As you commented, the actuation time must be less than 300 ms for AR glass applications. We measured the block force of the CASA actuating for 300 ms and compared it with the previous block force data measured for 2000 ms of actuation. In the case of 300 ms actuation, the peak force is measured to be approximately 0.5 N while the stroke is approximately 1.5 mm.

The actuation time is not the only variable that determines the block force. Applying a higher voltage and current also enabled the CASA to achieve 1.5 N of a peak force for 300 ms of actuation. The value of the voltage and current in this case was measured to be 2.5 V and 0.42 A respectively. We agree that presenting the block force for the actuation time of the application is important. However, presenting the maximum performance of the CASA can provide more useful information for readers to utilise actuators in many potential applications.

Fig. L1. Blocked force as a function of actuation stroke for 300 and 2000 ms actuation times.

Comment 9: In the “AR glasses prototype” subsection, the authors claim that “Although the actual human eye accommodation speed varies with age, it is generally 200~300 ms (5 Hz)⁴², which is sufficiently covered by CASA's actuation (~30 Hz)”. However, the actuation frequency in the AR glasses prototype presented in this paper is much lower. The actuation amplitude at 30 Hz is too small to be useful in any AR device. So this is not an appropriate claim.

Our response: We thank the reviewer for their comment. We used 30 Hz to emphasise the maximum frequency of the CASA. However, we agree that it is appropriate to describe a sufficiently high strain at the required frequency rather than only the maximum frequency at which the strain is very low. As shown in Fig. 2d, the displacement of the CASA at 5 Hz is approximately 1.5 mm which is sufficient for this application. We have also added the actuation of the AR glass prototype at a high frequency in Supplementary Video 6.

Revised manuscript:

(Page 24 line 17) *Although the human eye accommodation speed varies with age, it is generally in the range 200–300 ms (5 Hz)⁵⁷, which is sufficiently covered by CASA's actuation (approximately 1.5 mm of displacement under a frequency of 5 Hz).*

Comment 10: Sample length, actuation time, actuation frequency, actuation voltage and current of the SMAs embedded in CASA and externally used in the AR devices should be reported respectively in detail for the cyclic test in the “Cyclic test for temperature evaluation of SMA in CASA” subsection.

Our response: We added the information in Supplementary Table 6 (Comment 4) and referred to it in “Cyclic test for temperature evaluation of SMA in CASA” subsection.

Revised manuscript:

(Page 24 line 9) *The sample length, actuation time, applied voltage and current of the SMA wire are presented in Supplementary Table 6.*

Comment 11: A DMA test is highly recommended for the SMA actuators to find out the corresponding transition temperature (A_s , A_f , M_s and M_f), force and strain and optimize the operation condition correspondingly.

Our response: We thank the reviewer for the comment. We agree that a test to determine the corresponding parameters would improve the reliability of our results. As shown in Supplementary Fig. 9a, A_s , A_f , M_s , and M_f are 78, 86.1, 33.9, and 45.1 °C, respectively. In the martensite and austenite phases, we also measured the force and strain of the SMA wire to verify that our corresponding SMA model for optimisation is valid.

Revised manuscript:

Supplementary Figure 9. Thermomechanical properties of SMA wire. **a**, Differential scanning calorimetry (DSC) results showing transition temperature of SMA wire (A_s , A_f , M_s and M_f). **b**, Analytical model predictions and experimental results for force vs. strain relationship of the SMA wire. Red and black colour represent austenite and martensite phase, respectively.

(Supplementary Materials page 12) *Supplementary Figure 9. Thermomechanical properties of SMA wire. a. Differential scanning calorimetry (DSC) results showing transition temperature of SMA wire (A_s , A_f , M_s , and M_f). b. Analytical model predictions and experimental results for force vs. strain relationship of the SMA wire. Red and black colour represent austenite and martensite phase, respectively.*

Comment 12: A significant limitation of this work is that the actuation is bistable (on and off). Precise control of the strain inside the range is not presented, which limits the application of this actuator in AR application.

Our response: As the reviewer commented, the application of the actuator can be limited if it is only bistable. However, our actuator was designed for continuous actuation, with the possibility of position control. The CASA alone can consume a large amount of energy in applications where the actuated position needs to be maintained for a long time owing to the inherently low energy efficiency of the SMA. We developed the BPS to significantly reduce the input energy of the SMA actuator in applications with binary control. Thus, depending on the desired actuation mode, either the CASA alone or CASA with the BPS can be utilised for different application cases. CASA alone can be applied to applications such as haptic gloves, where various pressure ranges are required. The CASA with the BPS can be applied to other applications such as focus depth-switching AR glasses, where the actuated position needs to be maintained for a relatively long time.

Many studies have developed a position control system for an SMA-based actuator despite its non-linear characteristics of the SMA actuator. In this study, we focused on optimising the design process to achieve the desired and maximum actuator performance. We briefly show that the strain and force can be changed for different electric powers for user test of the haptic glove in Supplementary Fig. 17 and believe that precise position control of the continuous actuation of the CASA is also possible with feedback control in future work.

Revised manuscript:

Supplementary Figure 17. Applied force profile of haptic test. **a**, Force profiles (Off, A1, A2, A3, and A4) applied to participants in Test 1 (Actuation for 1 sec). **b**, Force profiles (Off, A1, A2, A3, and A4) applied to participants in Test 2 (Actuation for 20 ms).

Response to Reviewer #2 Comments

Comment 1: This manuscript addresses a really interesting topic of designing light-weight and high-power actuator based on Shape Memory Alloy (SMA) combined with strain-amplification. Overall, manuscript is well written and supported with supplementary materials including a number of video clips that demonstrate various aspects of the actuator. I am only able to provide review from my expertise related to on-body haptic feedback using SMA actuators. Recent work demonstrates that SMA actuators are particularly suited to physically deform the skin because of their large displacement length and high force-to-weight ratio (<https://dl.acm.org/doi/10.1145/3290605.3300718>). Would be good to cite this work.

Our response: We thank the reviewer for providing us with an additional reference to show the on-skin tactile interfaces with a large displacement length and high force-to-weight ratio of the SMA-based actuator. We have added the reference you suggested and modified the description in the manuscript to emphasise the advantages of the SMA-based actuator.

Revised manuscript:

(Page 3 line 2) *In addition to wearable optical devices, non-vibrating mechanotactile outputs have are important to generate natural and expressive tactile sensations on the skin through haptic devices¹⁷.*

Comment 2: Also, it would be good to understand the amount of expressivity (types of feedback) that can be generated by the proposed actuator or combination of multiple actuators. There was a very related work that looked into recreating various touch gestures using a matrix of wearable SMA patches (<https://dl.acm.org/doi/abs/10.1145/3313831.3376491>).

Our response: We thank the reviewer for for providing us with a reference to understand the expressivity of the haptic device. We can better express the advantages of the SMA-based actuator and CASA based on the reviewer's recommendation. We have added the following references and descriptions in the manuscript:

Revised manuscript:

(Page 3 line 4) *To convey the sensation of a large skin deformation, haptic devices require actuators with a high force-to-weight ratio and a large displacement. Combining multiple actuators in the limited area of the haptic device also enables more expressive tactile experiences¹⁸.*

Comment 3: In addition, the adaption of SMA actuators for haptic applications such as immersive VR is still at an early stage, especially because it requires VR developers to understand the complex non-linear behaviour of SMA-based actuators. Recent work, e.g. <https://dl.acm.org/doi/10.1145/3411764.3445613>, have proposed new approach that allows for easy fabrication of SMA based actuators on cloths. This could be a useful point to include in the discussion/future directions.

Our response: We thank the reviewer for providing us with a reference on on-body and wearable devices using SMA-based actuators. We have included a description of the platform for on-textile interfaces as future work:

Revised manuscript:

(Page 16 line 7) *In addition to wearable device applications, future work will attempt to develop a platform for on-textile interfaces⁵³ for clothing actuation keeping wearer comfort in mind.*

Response to Reviewer #3 Comments

Comment 1: This paper describes a type of SMA wire-driven actuator design for applications in augmented reality (AR). More specifically, a Compliant Amplified SMA actuator (CASA) is designed to amplify the small strain of SMA wire and a Bistable Parallelogram Linear State (BPM) is designed to further improve its energy efficiency. Two demonstrations are performed, one as a focus adjusting mechanism in a VR glass, and the other as a static force simulating mechanism worn on human's fingers. Overall, this paper is interesting and written in great detail. Some key parameters regarding actuators and devices are carefully investigated. The performance of the actuators is great and the supplementary videos show a very promising technology. However, the current version does not contain sufficient description of their novelty in terms of actuator design and applications. The following questions need to be addressed before considering publication in Nature Communications.

Our response: We thank the reviewer for the constructive review and comments. All comments raised by reviewer are addressed as follows. We hope the answers are clear and enough for understanding contribution of the paper.

Comment 2: The authors need to clearly define some of their key words earlier in their paper. For example, the "small form factor". How thick should the device be and how do we define a small form factor? Also, how fast should the focus adjusting system be? If it is to be done within 300 ms, it seems like the frequency of the actuator has no need to reach 30 Hz.

Our response: Thank you for your thoughtful comments. We agree that the expression 'small form factor' is not clear. Instead, the small form factor can be defined with reference to the typical size of other linear actuators that exhibit similar actuation performance.

A voice coil motor (VCM) is a linear actuator that is widely used in the auto-focusing module of smartphone cameras. To embed the actuator in a smart phone, the actuator needs to be designed to be as small and thin as possible while achieving a target stroke. There are commercial VCM actuators that have a stroke of approximately 3 mm, similar to that of our actuator. Specifications of the typical VCM actuator and CASA are shown below. The two commercial VCM actuators showed that the thickness should be much lower than at least 12 mm to be defined as a small form factor (low profile), while the actuation stroke is 3 mm.

Actuator	Stroke	Force	Weight	Area	Thickness
VCM #1	3.2 mm	4.4 N	7 g	15.9×15.9 mm ²	15.9 mm
VCM #2	3.2 mm	0.8 N	5.7 g	11.1×11.1 mm ²	12.7 mm
CASA without payload	3 mm	2.2 N	0.22 g	25×10 mm ²	5 mm
CASA under 80 g payload	4.5 mm				1 mm
NCC01-04-001-1X (VCM #1), HVCM-016-010-003-01 (VCM #2) were taken from datasheets of the H2W Technologies and the Moticont, United States.					

Revised manuscript:

(Page 4 line 22) *To surpass a CASA’s peak force of 2 N or stroke of 3 mm with a commercialized VCM, which is broadly employed in autofocus modules, the typical VCM actuator must be at least 70 times (>15 g) heavier or twice thicker (<12 mm) than the CASA.*

Comment 3: CASA actuator. The CASA mechanism proposed in this paper is similar to other Strain Amplification Mechanisms, both for SMA wire actuators and other types of actuators [1] [2]. The authors need to claim their uniqueness of their design to state CASA is one of their important contributions.

Our response: We thank the reviewer for critical comment. The main query is regarding is the novelty of our work compared to other studies. The strain-amplification technique is a well-known method that is widely used to develop piezoelectric and SMA actuators. Because our design is based on the fundamental mechanism of the strain-amplification technique, we understand that a similar schematic of the SMA actuator presented in the provided references could raise concerns about the novelty of our work.

Even though the actuators in our work and the references provided share the same fundamental strain-amplification mechanism, the differences in the material, structure of the actuators, and detailed amplification mechanism result in entirely different designs, modelling, performance, and applications.

Actuator	Stroke (Strain)	Blocked force	Weight	Size
CASA	3 mm (56%)	2.2 N	0.2 g	25×10×5 mm ²
Ref. 1	2 mm (28%)	N/A	N/A	50×7.2×7.2 mm ²
Ref. 2	2.5 mm (21%)	1.69 N	15 g	30×12×12 mm ²

Among the three actuators provided, the CASA has the highest strain, even with a similar or smaller actuator size. The highest strain of the CASA was achieved by optimising the design process and the structural differences. We established an optimisation model for combining the SMA wire and compliant beams to determine the optimal length of the SMA wire. Determining the optimal length of the SMA wire is critical for the practical usage of strain-amplification mechanism-based devices because it determines the actuation strain as well as the initial volume (height). We achieved the highest actuation strain under the given conditions with the derived pseudo-rigid-body model of the compliant beam and a one-dimensional nonlinear SMA model which are simple to use but precise and accurate for describing the characteristics of the actuator. In Ref. [1], the strain of the SMA itself is reported to be only 2%, which is not sufficiently stretched considering more than 3% of the strain of the SMA used in the CASA. Our model enables the user to determine the optimal length of the SMA to capitalise on the SMA to the maximum extent, resulting in high performance. Our actuator was also designed to minimise the use of additional components to reduce the weight (0.2 g). The CASA only consists of the SMA wire, small crimps, and two compliant beams that are precisely patterned to join each other to function as a hinge by themselves.

SMA actuators have an inherently low energy efficiency. Although power consumption is one of the most important issues for practical usage, it has not been sufficiently addressed in many SMA-based devices. We developed a bistable parallelogram linear stage (BPS) to compensate for low energy efficiency and tilting. These critical problems of SMA-based actuators are addressed by a single sheet of BPS which allows for a low-profile and compact form factor of the actuator.

In addition, we also exploited the resistance-changing characteristic of the SMA in the CASA to utilise it as a force sensor, which was not included in our previous manuscript. As shown in Figure 6, we employed CASA as a resistance-type force sensor to measure the external contact and added a sensing function to the haptic glove for tactile communication without embedding separate actuators and sensors in the glove. The details of the sensing capability are described in the revised manuscript and supplementary movie 8. We believe that the additional function of CASA supports the novelty of our study.

Revised manuscript:

(page 4 paragraph 2) *Previous studies^{21, 40} exploited the mechanism used to amplify the small strain of the SMA and the piezoelectric actuator. In contrast, we established an optimizing design of the CASA to achieve relatively high actuation strain (56.1%), power density (1.7 kW/kg), and thin form factor (5 mm of height) with the desirable force and frequency for practical use in AR wearable applications. The CASA also combines an SMA wire and a compliant structure that amplifies the strain without heavy rigid components required for amplification mechanism in the previous study.*

(page 5 line 13) *Last, we utilized the CASA as an resistance-type force sensor to detect external contact. The CASAs provide both sensing and actuation function without embedding separate actuators and sensors in the haptic devices.*

(Page 13-15) Sensing capability for tactile communication

In addition to actuation, the sensing function is highly desirable in wearable systems^{43, 44}. The integration of the actuation and sensing⁴⁵⁻⁴⁷ provides novel multifunctional devices increasing application potential for next-generation wearable systems. The CASA with the embedded smart material is advantageous for couple sensing and actuation for applications such as communication for deaf-blind people through the haptic glove.

The electrical resistance of the SMA can be varied by changing the strain. The SMA-based device can be controlled through resistance-based self-sensing techniques^{48, 49}. Figure 6a shows that the CASA can be utilized as a resistance-type force sensor for measuring the external contact. Before pressure is applied, the compliant bending beams of CASA and routed SMA wire with constant resistance (R_1) are in equilibrium. External pressure on the

CASA causes the bending compliant beams to straighten, resulting in the extension of the SMA wire. The resistance of the SMA wire in CASA, thus, increases (R2). Once the external pressure is removed, the compliant beams and the SMA wire are restored to their original states, and the resistance also returns to its original value (R1).

Figure 6b shows the loading and unloading characteristics of the two CASAs with different diameters of SMA wire (0.1 mm and 0.025 mm). The CASA with the 0.025 mm diameter SMA wire and 0.1 mm-thick compliant beams exhibits a relatively high sensitivity of 0.1498 N^{-1} under an applied force of 0.13 N. The CASA with the 0.1 mm diameter SMA wire and 0.2 mm-thick compliant beam exhibits a sensitivity of 0.01595 N^{-1} under an applied force of 1.1 N. The sensitivity and measurable force range are determined by the beam thickness and wire diameter. The high stiffness of the CASA with the thick beam (0.2 mm) and thick wire (0.1 mm) enables a wide range of measurable force, while the low stiffness of the CASA with the thin beam (0.1 mm) and thin wire (0.025 mm of diameter) provides high sensitivity. During unloading, the response characteristic aligns with the corresponding loading characteristic without noticeable hysteresis under the applied force. The results of the loading–unloading test conducted for approximately 1000 cycles demonstrate the durability of the CASA as a force sensor (Supplementary Fig. 20).

The CASA exhibiting sensing capability and high actuation performance can be used to develop highly compact haptic gloves for the tactile communication described in Fig. 6c without embedding separate actuators and sensors (Supplementary Movie 8). Tactile communication allows deaf-blind people to communicate through a Braille-based device⁵⁰⁻⁵². A Braille cell consisting of six dots that can represent letters, numbers, and special characters with the combination of raised dots⁵². Through the finger Braille communication method, the user can represent the six dots of Braille using the index, middle, and ring fingers of both hands⁵¹.

In Fig. 6c, users are shown wearing the haptic gloves with the CASAs embedded on the index (L3, R3), middle (L2, R2), and ring (L1, R1) fingers of both hands. User A presses the corresponding fingers on the ground to represent the Braille code as shown in Supplementary Fig. 21. The CASAs embedded on each finger are also applied by contact force by tapping fingers and the letter represented by the Braille code can be identified based on the resistance changes in the SMA wire in the CASAs. The resistance data of the CASAs

can be used to actuate the corresponding CASAs embedded in user B's haptic glove, allowing user B to receive the message. User B can send messages to user A in the same way.

Figures 6d and 6e shows the letters H, O, W, A, R, E, Y, O, U, ? and G, R, E, A, T, ! are typed by users by tapping the appropriate combination of the fingers on the desk with the haptic glove. The resistance changes in the CASAs as sensors are clearly identified for each letter. The previously demonstrated actuation characteristic of the CASA enables the user to sense received messages as well.

Fig. 6 | CASA with sensing capability and its application to tactile communication. a, Schematic of CASA’s sensing capability measuring the external contact. b, Normalized resistance as a function of applied force under loading–unloading test. c, Concept of tactile communication using actuation and sensing capability of the CASA. d and e, Normalized sensor output for each finger and corresponding Braille code to input H, O, W, A, R, E, Y, O, U, ? and G, R, E, A, T, !, respectively.

Supplementary Figure 20. Cyclic test results of loading–unloading loop for 5,000 s. a, 0.1 mm diameter of the SMA wire. b, 0.025 mm diameter of the SMA wire. c, Cyclic test results of 0.1 mm diameter of the SMA wire for 10 cycles d, Cyclic test results of 0.025 mm diameter of the SMA wire for 10 cycles. Cyclic test speed on (a) and (b) are 50 and 40 mm/min, respectively.

Supplementary Figure 21. CASAs embedded in the haptic glove for actuation and sensing capability. CASAs located in L1, L2, L3, R1, R2, and R3 are corresponded to each Braille code.

Comment 4: CASA actuator power density. The calculated power density of CASA is up to 1.7 kW/kg. This is definitely an amazing result. However, after further checking the displacement, force, as well as dynamic properties of CASA, the reviewer suspects this data might be incorrectly calculated. The authors need to recalculate it. It is true that the displacement and force, and thus energy density of CASA is great, yet its properties drop down quickly as frequencies increases. Thus, at high frequency like 30 Hz, both the force and the strain are small, ending up with a limited power density. If it is 1.7 kW/kg, the authors need to clearly describe at which frequency have they calculate the 1.7 kW/kg power density, as well the generated force and displacement related to this frequency on page 7. Also, the force vs frequency data need to be supplemented in Fig. 2d.

Our response: We apologise for providing insufficient information for calculating the maximum power density. For consistency in the calculation, we followed the process of calculating the maximum power density in the reference [25, a]. To obtain the maximum power density of the actuator, we applied a step input with a high voltage of 20 V for 20 ms under

various preloads. As the reviewer mentioned, the power density gradually decreased as the actuation frequency increased, which can be easily predicted by the relationship between the displacement and frequency plot in Fig. 2d. As shown in Fig. L2, the force applied by the CASA and the specific power are calculated as

$$p = \frac{1}{m_{CASA}} F_{CASA} \cdot v, \quad (10)$$

$$F_{CASA} = m_{Load} \cdot (a + g), \quad (11)$$

where F_{CASA} , m_{Load} , a , and g are the actuation force of the CASA, the mass of the applied load, acceleration of the mass, and gravitational acceleration, respectively. To calculate the power density, we show the force and velocity data of the actuator in Fig. L2.

25. N. Kellaris, V. G. Venkata, G. M. Smith, S. K. Mitchell, C. Keplinger, Peano-HASEL actuators: Muscle-mimetic, electrohydraulic transducers that linearly contract on activation. *Sci. Robot.* **3**, 1–11 (2018).

Reference for Revision

- a. Li, J., Sun, M. & Wu, Z. Design and Fabrication of a Low-Cost Silicone and Water-Based Soft Actuator with a High Load-to-Weight Ratio. *Soft Robotics* **8**, 448–461 (2021).

Fig. L2. Performance characterization of CASA. **a**, Schematic of experimental setup for measuring actuation stroke of the CASA with additional load. **b**, Actuation stroke of pre-loaded CASA when voltage is applied. t_i and t_e correspond to the times of initial and equilibrium actuation, respectively. **c**, Force plot of pre-loaded CASA. **d**, Velocity plot of pre-loaded CASA. **e**, Specific power plot of pre-loaded CASA

Comment 5: How CASA and BPS are combined should be described.

Our response: We apologise for providing insufficient information regarding the fabrication procedure of the actuator module for AR glass devices. We have added an explanation of the reviewer's comments in the Methods section.

Revised manuscript:

(page 18 paragraph 2) *The CASA embedded in the actuator module is composed of two compliant beams, as shown in Supplementary Figure. 10c. One beam has a connecting hole, which is combined with a lower part in Supplementary Fig. 10a. The CASA combined with the lower part is bolted to the external frame. The counter SMA wire passes through holes located in the centre of the BPS. Both sides of the counter SMA wire are clamped and anchored at the external frame.*

Comment 6: The demonstrations of both the focus adjusting and the static force simulating lack quantitative evaluation to support the superiority of their design. The authors claim their actuators are of high speed, then the time response of their final integrated device should be clearly shown. Also, how does the user feel about wearing the AR glass with such a focus adjusting mechanism? Does it really release fatigue for the users? These need to be quantitatively evaluated. Human tests need to be done to verify the functionality of the haptic glove design as well, as sometimes, giving some amount of force on the hand, does not necessarily mean the wearers feel like they are touching something. The reviewers believe more in-depth evaluation of the two haptic devices need to be conducted, to prove the superiority of their proposed device.

Our response: We regret providing insufficient information to prove the quantitative evaluation of the actuator and integrated devices. We have added an additional explanation and have performed user tests on human volunteers. User tests were conducted with the help of human volunteers complying with an approved IRB protocol (IRB No. 202203-HB-EX-001).

To verify the functionality of the haptic glove, we conducted user tests in which the haptic glove conveyed a gradually increasing contact force for 1 s (Test 1) and highly impulsive pressure for 20 ms (Test 2). For the quantitative test, we recruited 12 volunteers (three females and nine males, ranging in age from 25 to 32 years). Participants wore a haptic glove on their left hand, and the CASA on the index fingertip was only actuated to convey the sensation. Supplementary Figure 17 shows the applied force profiles in Tests 1 and 2. We have added the experimental results, as shown in Fig. 5 and Supplementary Fig. 17 and 18, and wrote a description in the revised manuscript.

The time response of our final integrated AR device is shown in Supplementary Fig. 16c. Regarding the fatigue caused by VAC (Vergence-Accommodation Conflict), it is a well-known issue and is the main cause of visual fatigue in spectacler AR glasses. However, visual fatigue may be perceived differently according to age, sex, and neurological and psychological factors [16]. Attempts have been made to evaluate this quantitatively [4, 6]. In this study, to mitigate VAC, we propose a method to implement binary depth of a virtual image by applying CASA as a display actuating element for AR glasses. The ideal solution to allow AR glasses wearers

to experience approximately natural viewing should be able to match the vergence and accommodation of the eye in depth. A solution capable of realising continuous depth using a varifocal lens has been proposed [12], but an AR device using a varifocal lens becomes bulky and heavy.

Based on the previous study about ‘Zone of Comfort’ [4], the results for improving VAC with binary depth switching of virtual image was provided. In this study, we propose a solution for binary switching of a display by applying CASA. The representative depths of the image are divided into far and near depths, and the boundary value between the far and near depths is set to 2 D (dioptr). This is based on the arm length of a general person. When the representative depth of the main object observed by the wearer crosses the boundary value, CASA is activated. The slow actuating speed of CASA can act as a factor that allows the wearer to perceive the awkwardness of image conversion. Based on the dynamic response research of the eye [57], the CASA’s switching time, ‘300 ms’, was similar to the value of the dynamic accommodation of the eye measured in a previous study. Therefore, the wearer does not feel awkward based on the transition of the virtual image.

4. Shibata T, Kim J, Hoffman DM, Banks MS. The zone of comfort: Predicting visual discomfort with stereo displays. *Journal of vision* **11**, 11 (2011).
6. Padmanaban N, Konrad R, Stramer T, Cooper EA, Wetzstein G. Optimizing virtual reality for all users through gaze-contingent and adaptive focus displays. *Proceedings of the National Academy of Sciences* **114**, 2183-2188 (2017).
12. Wilson A, Hua H. Design and demonstration of a vari-focal optical see-through head-mounted display using freeform Alvarez lenses. *Optics express* **27**, 15627-15637 (2019).
16. Kramida G. Resolving the vergence-accommodation conflict in head-mounted displays. *IEEE transactions on visualization and computer graphics* **22**, 1912-1931 (2015).
57. Campbell FW, Westheimer G. Dynamics of accommodation responses of the human eye. *J Physiol* **151**, 285-295 (1960).

Revised manuscript:

(page 12 line 15) *The haptic glove can, thus, convey the highly impulsive pressure as well as gradually increasing and static pressure to the user.*

(Page 12 paragraph 3) *For quantitative tests, we recruited 12 volunteers (3 females and 9 males, ranging in age from 25 to 32 years old). They wore the haptic glove on the left hand, and the CASA, worn on the index fingertip, was only actuated to convey the sensation as shown in Supplementary Fig. 17. Two types of tests were performed to measure the signal intensity perceived by the users for different magnitudes of the contact forces. The user perceived gradually increasing contact forces generated for 1 s in Test 1 and impulsive forces generated for 20 ms in Test 2 as shown in Supplementary Fig. 17. The generated peak forces of 4 different magnitudes were ranged from 0.083 to 2.052 N in Test 1 and from 0.045 to 1.135 N in Test 2. Before the test started, each user was shown all stimuli once and told the corresponding stimulus number for each (A1, A2, A3, and A4). The user then randomly perceived one of four different magnitudes of the forces or the 'off' state in each test and were asked to identify the stimulus. Each contact force and off state were randomly applied for 10 times (50 stimuli in total for each test).*

(Page 13 paragraph 2, 3, and 4) *Figure 5e shows the confusion matrix, in which the rows and columns corresponds to the applied signal and signal identified by the user in Test 1, respectively. The users have a high Identification Rate (IR) of over 80% for the lowest and highest magnitudes of forces (A1 and A4). For A2 and A3, the users have an IR of over 55%. After experiencing many series of stimuli, the users are unsure about identifying the specific signal, especially the two middle stimuli (A2 and A3). However, the averaged intensity of each signal in Fig. 5f and Supplementary Fig. 18 show that the users are able to compare the different intensity during the test in overall.*

Figure 5g and 5h and Supplementary Fig. 19 show corresponding perception results of Test 2 measuring how users perceive impulsive forces in an instant (20 ms). The users correctly distinguish the off state and the lowest magnitude of sudden force with a high IR of over 90%. As shown in Supplementary Fig. 17b the differences of peak force in Test 2 are comparatively low. The differences were ranged from approximately 0.13 to 0.71 N. Due to the relatively low difference and actuation time, the users have an IR between 40 and 55%. Similar to Test 1, the averaged intensity in Fig. 5h shows the users are able to compare the intensity of the signal overall.

The perception results demonstrate that the CASA is able to convey various sensation, such as large skin deformation (2 N) and gentle touch (0.05 N), at different actuation time.

Fig. 5 | Actuation of CASA in haptic glove. **a**, (top) Conceptual illustration of interaction between human and virtual image in AR using haptic gloves with CASA. (bottom) Actuation mechanism of the CASA in a haptic glove for realizing tactile sensation. **b**, (left) Force–displacement relationship of the CASA in the fingertip of the haptic glove. (right) Pressure transmission mechanism visualized with the transparent haptic glove integrated with CASA on the fingertip. **c**, High power actuation of thin haptic glove integrated with CASA throwing 10 g of the weight. **d**, Thin haptic glove integrated with multiple CASAs minimizing interference during various motions of fingers. The red dashed circles indicate CASAs embedded in the haptic glove. **e**, The confusion matrix of Test 1, showing Identification Rate (IR) of 12 untrained volunteers **f**, Average reported feeling of users (scale 0 to 4) versus Applied stimuli in Test 1. **g**, The confusion matrix of Test 2, showing IR of 12 untrained volunteers **h**, Average reported feeling of users (scale 0 to 4) versus applied stimuli in Test 2.

Supplementary Figure 17. Applied force profile of haptic test. a, Force profiles (Off, A1, A2, A3, and A4) applied to participants in Test 1 (actuated for 1 s). **b,** Force profiles (Off, A1, A2, A3, and A4) applied to participants in Test 2 (actuated for 20 ms).

Supplementary Figure 18. Test 1 results. Average reported feeling of user vs. applied stimuli (Test 1) for all 12 users.

Supplementary Figure 19. Test 2 results. Average reported feeling of user vs. applied stimuli (Test 2) for all 12 users.

REVIEWERS' COMMENTS

Reviewer #1 (Remarks to the Author):

The authors address all my concerns and questions appropriately. My last suggestion is to include more works regarding the SMA actuator and compare the power density. One example is "Coordinated Use of Structure-Integrated Bistable Actuation Modules for Agile Locomotion" by Nishikawa et al.

Reviewer #3 (Remarks to the Author):

I have carefully read the authors' responses to my proposed concerns and I think they have made great effort to address those concerns. Also, I feel their revisions and answers are sufficient to address the concerns. I support this manuscript to be accepted in Nature Communications.

Response to Reviewer #1 Comments

Comment 1: The authors address all my concerns and questions appropriately. My last suggestion is to include more works regarding the SMA actuator and compare the power density. One example is "Coordinated Use of Structure-Integrated Bistable Actuation Modules for Agile Locomotion" by Nishikawa et al.

Our response: We thank the reviewer for the constructive comments. We added the reference and the quality of our paper has greatly improved thanks to the critical comments the reviewer provided.

Response to Reviewer #3 Comments

Comment 1: I have carefully read the authors' responses to my proposed concerns and I think they have made great effort to address those concerns. Also, I feel their revisions and answers are sufficient to address the concerns. I support this manuscript to be accepted in Nature Communications.

Our response: We thank the reviewer for the constructive comments. The quantitative evaluation section for our haptic devices has greatly improved thanks to the critical comments the reviewer provided.